# Design and Analysis of a Wearable Upper Limb Rehabilitation Robot with Characteristics of Tension Mechanism

**Zaixiang Pang [1,2], Tongyu Wang [1,*], Zhanli Wang [2], Junzhi Yu [3], Zhongbo Sun [4] and Shuai Liu [2]**

[1] School of Mechatronical Engineering, Changchun University of Science and Technology, Changchun 130022, China; pangzaixiang@ccut.edu.cn

[2] School of Mechatronical Engineering, Changchun University of Technology, Changchun 130012, China; wangzl@ccut.edu.cn (Z.W.); m15526835521@163.com (S.L.)

[3] State Key Laboratory for Turbulence and Complex Systems, Department of Mechanics and Engineering Science, BIC-ESAT, College of Engineering, Peking University, Beijing 100871, China; yujunzhi@pku.edu.cn

[4] College of Electrical and Electronic Engineering, Changchun University of Technology, Changchun 130012, China; zbsun@ccut.edu.cn

* Correspondence: wty@cust.edu.cn; Tel.: +86-1394-482-3099

**Abstract:** Nowadays, patients with mild and moderate upper limb paralysis caused by cerebral apoplexy are uncomfortable with autonomous rehabilitation. In this paper, according to the "rope + toothed belt" generalized rope drive design scheme, we design a utility model for a wearable upper limb rehabilitation robot with a tension mechanism. Owing to study of the human upper extremity anatomy, movement mechanisms, and the ranges of motion, it can determine the range of motion angles of the human arm joints, and design the shoulder joint, elbow joint, and wrist joint separately under the principle of ensuring the minimum driving torque. Then, the kinematics, workspace and dynamics analysis of each structure are performed. Finally, the control system of the rehabilitation robot is designed. The experimental results show that the structure is convenient to wear on the human body, and the robot's freedom of movement matches well with the freedom of movement of the human body. It can effectively support and traction the front and rear arms of the affected limb, and accurately transmit the applied traction force to the upper limb of the joints. The rationality of the wearable upper limb rehabilitation robot design is verified, which can help patients achieve rehabilitation training and provide an effective rehabilitation equipment for patients with hemiplegia caused by stroke.

**Keywords:** rehabilitation robot; characteristics of tension mechanism; wearable; upper limb; design and analysis

## 1. Introduction

The number of young patients with functional impairment of the upper limbs caused by stroke has increased rapidly, as influenced by accelerated pace of life, poor lifestyles and environmental factors [1,2]. Limb movement disorder, which is caused by hemiplegia after stroke, not only reduces the quality of life of patients, but also brings great pain to their physiology and psychology. Effective rehabilitation training can improve the defect of patients' nerve function and maintain the degree of joint activity; it also prevents joint spasms and enhances the final rehabilitation degree of patients' motor functions significantly [3]. The traditional rehabilitation training is one-to-one auxiliary exercise for patients by therapists. This method is difficult to develop an effective treatment plan, and it is tough to control accurately [4]. With the development of rehabilitation robot technology and rehabilitation

medicine, the rehabilitation robot has become a novel motor nerve rehabilitation treatment technology. It is of great significance to take advantage of rehabilitation robot technology for rehabilitation training to the recovery of limb function of stroke patients [5]. The traditional methods of treatment, which are based on the therapist's clinical experience, have the problems of large staff consumption, long rehabilitation cycles, limited rehabilitation effects, and so on. The research and application of rehabilitation robot system is expected to alleviate the contradiction between supply and demand of rehabilitation medical resources effectively, and improve the quality of life of stroke patients [6,7].

The upper limb rehabilitation robot can be divided into two types according to the structural form: terminal traction and exoskeleton type. The terminal traction type mainly provides the rehabilitation training of plane movement. However, the exoskeleton type extends the rehabilitation training range from plane to three-dimensional (3D) space, which can assist the affected limb to complete the rehabilitation training in 3D space. The exoskeleton rehabilitation robot generally drives the movement of the patient's limbs through the auxiliary device (also known as exoskeleton mechanical structure). The structure of the auxiliary device is similar to the skeleton structure of the human limbs. During the training, the patient's limbs and the corresponding parts of the auxiliary device are bound together, and the connecting rod of the auxiliary device swings around the corresponding joint, so as to bring the moving limbs into motion. It can make the patient's limbs train in different postures through controlling the trajectories of power-assist device. At present, the structural design method of exoskeleton rehabilitation robot is one of the hot issues in the research of rehabilitation robot. Owing to different mechanical structures and rehabilitation principles, a variety of exoskeleton rehabilitation robots are developed, e.g., a dynamic exoskeleton system ADEN-7 robot with 7 degrees of freedom [8], an ARMIN robot with six degrees of freedom (four active and two passive) semi exoskeleton structure [9], an ARMEO robot providing arm weight reduction support system training, enhancing performance feedback and evaluation tools [10], etc. In addition, the pneumatic muscle is used as a driver to realize four degrees of freedom active auxiliary motion RUPERT robot [11], hydraulic drive robot LIMPACT [12], suspended rope drive robot CAREX [13]. After that, researchers developed and designed the upper limb rehabilitation robot based on pneumatic muscle drive, unpowered upper limb rehabilitation robot, hybrid drive upper limb rehabilitation robot and under drive exoskeleton upper limb rehabilitation robot [14–22]. The exoskeleton rehabilitation robot solves the problem of controlling the motion amplitude and moment of each joint of human body in the process of rehabilitation training, and overcomes the disadvantage that the end guided rehabilitation robot can only perform simple rehabilitation training (linear motion or circular motion) with small motion amplitude. Currently it is a relatively safe and efficient rehabilitation robot structure. However, in the design of exoskeleton prostheses, the matching of mechanical joint motion axis and human joint motion axis is very important. The exoskeleton produces unexpected forces at the patient's joint under mismatched condition, which not only causes joint pain and injury to the patient, but also limits the movement space of the patient's limbs, and reduces the effect of rehabilitation training. Therefore, the axis of each pair of motion is matched with the rotation center of each joint of the human body as far as possible in the design of exoskeleton rehabilitation apparatus. The motion of each joint of exoskeleton rehabilitation device is realized mainly by rotating or moving the pair, and good results have been obtained [23,24]. Compared with the artificial rehabilitation treatment, the rehabilitation robot system has the advantages of high training accuracy, easy to quantify the amount of exercise, and long-term one-to-one scientific rehabilitation treatment for patients.

To satisfy the rehabilitation needs of patients with limb disorders, a wearable upper limb rehabilitation robot is designed and developed in this article, which is mainly a device for mid-term semi-active rehabilitation training and post-active rehabilitation training for stroke patients. Owing to understanding the disadvantages of traditional rehabilitation training and the performances of rehabilitation robots, combined with the human upper limb muscle anatomy characteristic and relevant parameters, we determine the arm movement of each joint angle range from all the bones and joints of upper limb movement characteristics, this paper proposes a design scheme of the tensegrity structure

wearable upper limbs rehabilitation robot. The wearable upper limb rehabilitation robot is utilized to the exercise rehabilitation treatment of hemiplegic limb to maintain the range of motion of the limb, prevent the muscle atrophy of the limb, enhance the muscle strength of the limb, and promote the recovery of the limb function. Therefore, it can provide an effective rehabilitation equipment for patients with hemiplegia of upper limb caused by stroke.

In this paper, due to the study of anatomy, motion mechanism and motion range of human upper limb, the motion angle range of each joint is determined for human arm, and the mechanical mechanism on each degree of freedom is designed for wearable upper limb rehabilitation robot. First, to establish the spatial pose relationship between each motion component and the end-effector of the wearable upper limb rehabilitation robot, the motion model is established with the Denavit–Hartenberg (D-H) parameter method and the motion space is analyzed for wearable upper limb rehabilitation robot. The kinematics analysis is used to analyze the motion of the wearable upper limb rehabilitation robot. Secondly, to verify whether the wearable upper limb rehabilitation robot can realize the auxiliary upper limb functional rehabilitation training, the working space is analyzed for the wearable upper limb rehabilitation robot. Thirdly, to analyze the output torque of wearable upper limb rehabilitation robot, the dynamic simulation of the robot is carried out. Lastly, the control system of wearable upper limb rehabilitation robot is designed, which obtained the tracking results of robot rehabilitation training. It further verifies that the rationality of the design of wearable upper limb rehabilitation robot.

The main contributions of this paper are summarized as follows:

(1) Owing to the anatomy theory, motion mechanism and range of human upper limbs, a novel wearable upper limb rehabilitation robot with tension mechanism is firstly designed, investigated and analyzed for upper limb injured patients based on flexible transmission during rehabilitation training process. A cable-driven modular parallel joints are innovatively designed for elbow/wrist and a shoulder joint driven by a toothed belt. All the cable-driven motors are rear-mounted to achieve long-distance transmission and reduce the drive inertia of the end joints. The gear belt is exploited to drive the joints of a wearable upper limb rehabilitation robot, which realizing high precision meshing. The design approach of the wearable upper limb rehabilitation robot facilitates the rehabilitation training of the joint, effectively reduces the volume, mass and inertia of the actuators, and achieves the lightweight design of the overall structure.

(2) Additionally, this paper proposes a flexibly parallel mechanism of humanoid wrist driven by rope and supported through a compression spring. The fixed base and moving platform of the wearable upper limb rehabilitation robot are connected by three ropes and a conical compression spring. The springs are designed by simulating the human wrist and support the mobile platform to complete the wrist movement, while the ropes are constructed via simulating the wrist muscles to control the wearable upper limb rehabilitation robot. In this paper, the design approach will contribute to the further study of parallel mechanisms with flexible joints. The results will play an important role in reappearing the movement of human wrist and promote the development of rehabilitation robot and rope drive technology.

(3) The kinematics and workspace of the wearable upper limb rehabilitation robot are verified and analyzed based on the D-H method and Monte Carlo method. It demonstrates that the wearable upper limb rehabilitation robot can satisfy the requirements of rehabilitation training through kinematics/dynamics analysis and rehabilitation training experiments. Therefore, it also further verifies that the feasibility and effectiveness of the design method, which provides a valuable idea for improving rehabilitation robot mechanism.

The rest of this paper are organized as follows. From the perspective of bionics, Section 2 analyzes the joints of the wearable upper limb rehabilitation robot, and designs the mechanical system model of the joints. To obtain the relationship of rotation and translation between adjacent members of the wearable upper limb rehabilitation robot and its terminal pose, the kinematics model is investigated and analyzed for the wearable upper limb rehabilitation robot, and the correctness of the solutions is demonstrated for forward/inverse kinematics of the wearable upper limb rehabilitation robot in

Section 3. Owing to the Monte Carlo method, the workspace and full workspace are analyzed and obtained for the wearable upper limb rehabilitation robot in Section 3. Section 4 simulates and analyzes the kinematics/dynamics model of the wearable upper limb rehabilitation robot to demonstrating the stability of the motion state and verifying the output torque of the motor can satisfy the needs of rehabilitation training process. In Section 5, the control system is designed for the wearable upper limb rehabilitation robot, and the rehabilitation training process is completed for upper limb injured patients, followed by the tracking result of the robot rehabilitation training. Conclusions are drawn in Section 6.

## 2. Structure Design of the Wearable Upper Limb Rehabilitation Robot

### 2.1. Shoulder Joint Design

From the anatomical point of view, the upper limb of human body mainly includes three joints: shoulder, elbow and wrist [25,26]. The shoulder joint is the most flexible ball and socket joint in the whole body, which can perform flexion/extension, internal rotation/external rotation, abduction/adduction and other movements. The maximum range of flexion/extension angle of shoulder joint is 135°, abduction/adduction is 135°, internal rotation/external rotation is 110°, and the range of motion is largee [27]. At the same time, since the three rotating pairs are at the beginning of the human upper limb motion chain, the operation space of the terminal effector (hand) is most affected by the motion of the joint at the beginning of the motion chain. Considering that any one degree of freedom will cause the hand to fail to complete some routine actions, the three degrees of freedom of the shoulder joint are reserved [28]. The shoulder joint is composed of the glenoid of the humeral head and scapula, which is connected by the joint capsule. It belongs to the ball and socket joint. It rotates around three mutually vertical axes respectively. At the same time, the shoulder joint is simplified as a connecting rod to connect the shoulder joint and the elbow joint. The degrees of freedom of shoulder joint motion is illustrated in Figure 1.

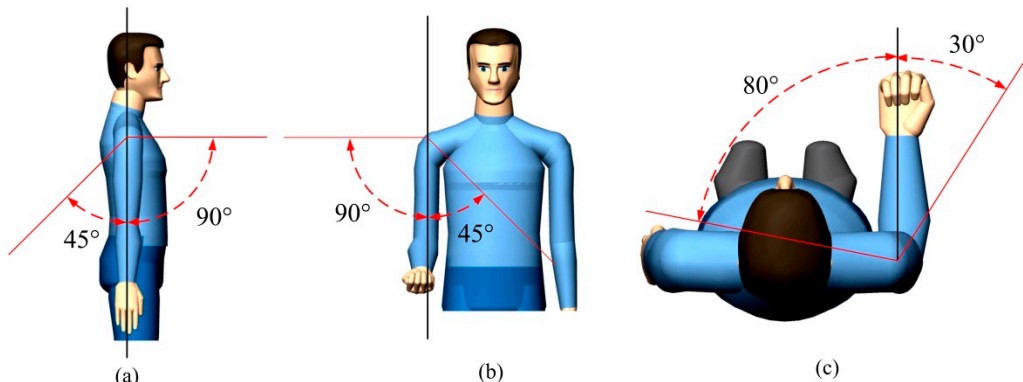

**Figure 1.** Shoulder joint freedom of motion. (**a**) Flexion/extension; (**b**) abduction/adduction; (**c**) internal rotation/external rotation.

The overall structure of the rehabilitation robot depends on the movement of the shoulder joint and the characteristics of different body shapes. The structure of the wearable upper limb rehabilitation robot shoulder joint is depicted in Figure 2. There are three degrees of freedom in the shoulder structure, in which the flexion/extension and abduction/adduction of the shoulder joint are driven and connected by harmonic motor. The in-swing/out-swing degree of freedom with the arm as the axis cannot be used due to the need to wear. The arc-shaped rack on the arc-shaped slide rail is meshed with the gear of the output shaft of the reducer, and the servo motor is used as a driving force to transmit the movement of the input shaft of the reducer to the arc-shaped rack, so as to realize the internal/external rotation of the shoulder joint. In order to ensure a certain motion accuracy, two sets of pulley blocks are installed on both sides of the arc-shaped rack. The function is to restrict the movement of the arc-shaped guide

rail along the direction of the arc, and play the role of limit. Since the position of the shoulder joint internal/external rotation mechanism is far away from the base, the quality of the mechanism will have a great impact on the control during the movement. After repeated experiments, it is determined that the arc joint rack is used to achieve the shoulder joint internal/external rotation movement. Because the curved rack needs to cooperate with the pulley set, the driving gear is not allowed to directly mesh with the curved rack in the assembly space. To this end, we have added a passive gear between the active gear and the curved rack, which can reduce the output torque of the motor reduces the quality of the mechanism and the structure is more compact and beautiful.

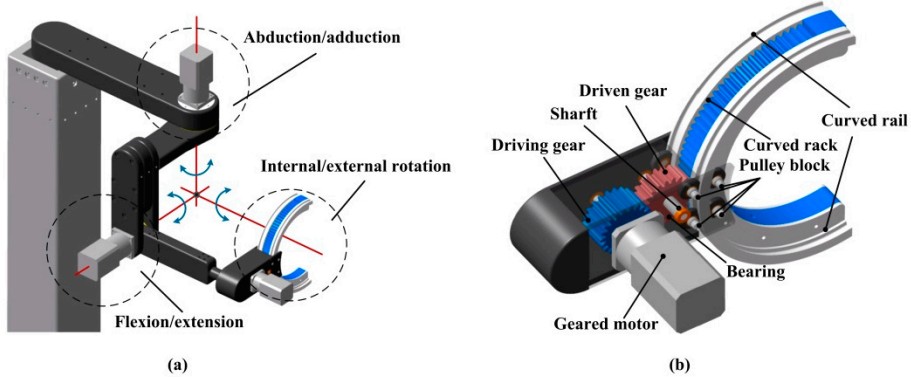

**Figure 2.** The structure diagram of shoulder joint. (**a**) Three-dimensional (3D) model of shoulder joint; (**b**) 3D model of shoulder joint rotation degree of freedom.

The transmission mechanism of the shoulder joint internal/external rotation mechanism is active gear-passive gear-arc rack, where both ends of the arc racks are provided with shoulders. The rack gear meshes. Once it exceeds the rehabilitation range, the passive gear will be blocked by the shoulder and cannot continue to move, which ensuring the safety of the patient and avoiding secondary injuries to the patient.

The structure uses gear transmission instead of belt transmission, with high transmission accuracy, fast response speed, stable transmission and strong bearing capacity. The number of teeth of passive gear and arc-shaped rack can limit the movement range of patient's shoulder joint rotation and ensure the safety of patients. Above all, it satisfies the requirements of high-precision rehabilitation training.

## 2.2. Elbow Joint Design

The elbow joint is the most important joint connecting the upper arm and forearm. It is a main trochlear joint. The joint movement is realized by the relative movement of cartilage. The flexion and extension of the upper limb depend on the joint, including flexion/extension and internal rotation/external rotation of the forearm [29]. These two degrees of freedom in the human body when the upper limb completes the daily activities of the movement angle: elbow joint flexion/extension of the maximum angle range is 135°, forearm internal rotation/external rotation is 90°, the largest range of motion should be considered, the elbow joint mainly completes such as eating, holding things, touching the head and so on in people's daily life. If the elbow joint movement is restricted, it will be greatly restricted, and other joints, which will also have a greater impact for the patient's daily life. Therefore, the elbow joint plays an important role in the upper limb joint. However, when the current arm is fixed internal/external rotation, there is almost no internal/external rotation of shoulder joint [30]. Meanwhile, based on the redundancy of the freedom of the motion mechanism, the above analysis of the shoulder joint has already considered the internal/external rotation movement. To facilitate the control and the stability of the structure, it can be considered that the internal/external rotation of the forearm is fixed to move together, so that the internal/external rotation of the shoulder is designed to the maximum angle range of 110°. As a result, the flexion/extension of elbow joint is equivalent to a single joint motion pair. The degree of freedom of elbow joint movement is shown in Figure 3a.

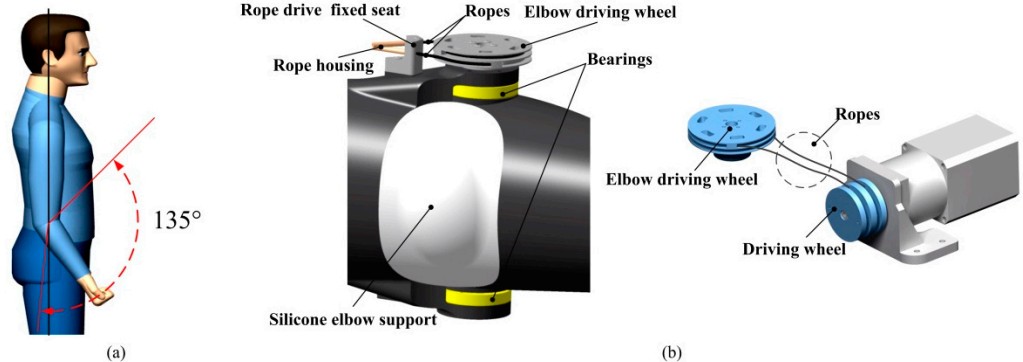

**Figure 3.** Elbow joint structure diagram. (**a**) Elbow joint freedom of motion; (**b**) 3D model of elbow joint.

As illustrated in Figure 3b, the elbow motion mechanism is constructed by two-way winding coil structure. It avoids the synchronization of double motor winding mechanism by two-way precise drive of the motor. The driven part of the elbow joint movement mechanism is mounted on the base, and the two-way driven plate of the motor transmits the power to the two-way wire plate of the elbow through the rope, thus, it completes the elbow flexion/extension motion. Since the elbow needs to bear a large torque and has excellent rotational accuracy, two crossed roller bearings are applied to the elbow joint. As a consequence, it increases the bearing capacity of the elbow mechanism, and reduces the radial error of the rotating shaft. The use of the silicone pad increases the wearing comfort and aesthetics of the device, which satisfies the requirements of the elbow movement when the human arm moves and also makes it more comfortable to wear.

*2.3. Wrist Joint Design*

The wrist joint is a small joint of human body, which is mainly used to connect the forearm and hand, and consists of a palm and fingers. The wrist joint has two degrees of freedom to complete flexion/extension, abduction/adduction. When these two degrees of freedom are combined with pronation and supination around the long axis of forearm, the wrist joint increases the third degree of freedom (passive flexion/extension and extension) [31,32]. During the rehabilitation training process, the traction of the wrist joint does not need a lot of force. In the meanwhile, the amplitude of passive flexion, extension and extension of wrist in the three degrees of freedom is small. The wrist can be fully trained through the other two degrees of freedom currently. Therefore, in order to reduce the complexity of the mechanical structure and restore the wrist joint of the human upper limb to the maximum extent, and simplify the structure and reduce the control difficulty on the premise of ensuring the basic functions, the passive flexion and extension and extension are not included in the design requirements of the wearable upper limb rehabilitation robot described in this paper. When the upper limb of human body completes daily activities, the maximum range of flexion/extension angle is 150°, and the abduction/adduction is 50°. The wrist joint can be equivalent to a spherical hinge mechanism. Based on the inherent rigidity of the general mechanical structure, and better map of the movement structure of the human wrist, the wrist of the wearable upper limb rehabilitation robot is designed with a flexible structure tower spring. In this paper, people mainly complete such actions as eating, taking things and touching their heads in the daily life. The wrist joint not only has a high frequency of motion, but also is the part of the upper limb that bears the largest load in the process of supporting, pushing and pulling. The design of wearable upper limbs rehabilitation robot mainly for medium-term and semi-active rehabilitation training in patients with cerebral apoplexy and late active rehabilitation training device, furthermore, the wrist of patients should have certain activity. During rehabilitation training, patients need to hold the end adjusting grip of the wearable upper limb rehabilitation robot, and the upper limb follows the robot to do corresponding rehabilitation training. Figure 4a shows the degree of freedom of wrist movement.

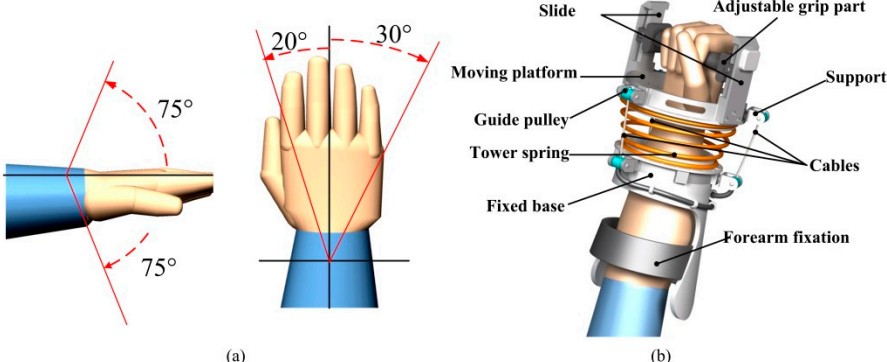

**Figure 4.** Wrist joint structure diagram. (**a**) Wrist joint freedom of motion; (**b**) 3D model of wrist joint.

As shown in Figure 4b, a flexible parallel mechanism is proposed to simulate the human wrist with a rope drive. The wrist adopts the hand-wrist-forearm connection. The front and rear sections of the wrist are connected by a tapered compression spring, which is used to simulate the motion of a wrist joint. There are three sets of rope mechanism around, each set of rope mechanism is separated by 120° to simulate the wrist muscles, which complete the drive and control of the wrist. In addition, control mechanism is equipped with a power source to be placed in the base part.

Owing to load and deformation are nonlinear, and comparing with a cylindrical helical spring, the conical helical spring has a greater stability and prevents resonance phenomenon, which is applied more and more widely. Specifically, if the load does not make the spring coil contact, the relationship of the load and deformation is linear, and if the load continues to increase, then the spring contacts from a large ring, and the relationship of load and deformation is nonlinear.

The flexible parallel mechanism takes the human wrist as the bionic object, where the fixed ring is equivalent to the radius and ulnar complex, and the moving ring is equivalent to the metacarpal bone. The driving rope and spring represent the muscles and ligaments around the wrist respectively, which providing kinetic energy and support for the motion of the radial and middle wrist joints. The parallel mechanism uses three servo motors to drive three ropes, which realize the wrist flexion/extension and ulnar/radial movement of the robot. It can not only reduce the flexible degree of freedom, but also enhance the stability of the mechanism, and make the mechanism satisfy the motion amplitude of the wrist under different angles when the mechanism is in retraction and abduction, flexing and stretching. Therefore, the mechanism can achieve the wrist joint adduction and abduction, flexion straight action. The flexible wrist joint driven by rope is mainly composed of three parts: adjustable grip, flexible parallel mechanism and forearm fixation.

According to above analyses of the motion freedom of each joint of the upper limb, a six degrees of freedom wearable upper limb rehabilitation robot with the characteristics of tension mechanisms is proposed in this paper. The six degrees of freedom of the rehabilitation robot include three degrees of freedom of the shoulder joint: flexion/extension, internal/external rotation, abduction/adduction; one degree of freedom of the elbow joint: flexion/extension; one degree of freedom of the forearm: internal/external rotation; two degrees of freedom of the wrist joint: flexion/extension and abduction/adduction. The 3D structure of the rehabilitation robot is shown in Figure 5. The wearable upper limb rehabilitation robot designed in this article is wearable, which consists of a vertical wearable rehabilitation robot and a seat: the patient sits on the seat during the rehabilitation process; the patient's arm passes through the traction of the rehabilitation robot institutional contacts. The arm passively performs full-circle rotation by driving the curved rack and provides resistance during active movements of the shoulders. The seat height adjustment mechanism can be adjusted according to the height of the upper limb and the body shape of the human body, adapt to different treatment environment and the difference of the patient's body shape, and ensure that the affected limb is trained on the sagittal plane during the rehabilitation process.

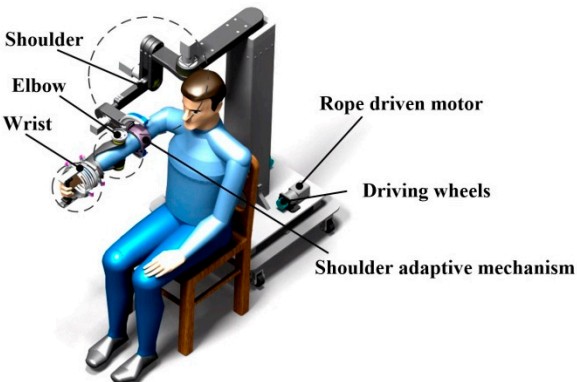

**Figure 5.** A 3D model of the wearable upper limb rehabilitation robot.

Each joint mechanism of the wearable upper limb rehabilitation robot is closely combined with the upper arm, forearm and wrist of the human body, and the rehabilitation training of multiple joints of the upper limb is realized by using the freedom of the rehabilitation robot hinge type rotating pair, arc rack, tower spring, etc. On the sagittal plane, the rehabilitation robot can realize the flexion/extension of the shoulder 0°–100°, abduction/adduction 0°–120°, internal/external rotation out 0°–110°; the flexion/extension of the elbow 0°–105°, internal/external rotation out 0°–90°; the flexion/extension 0°–90° joint motion analysis angle range of the wrist meets the angle requirements of the upper limb rehabilitation training process [33]. The free distribution of human upper limbs and rehabilitation robot degrees of freedom are shown in Table 1.

**Table 1.** Free distribution of human upper limbs and robot degrees of freedom.

| Parts | Degrees of Freedom | Movement Range of Human | Movement Range of Robot [34,35] |
|---|---|---|---|
| Shoulder | Flexion/extension | 0°–90°/0°–45° | 0°–90°/0°–10° |
| Shoulder | Abduction/adduction | 0°–90°/0°–45° | 0°–75°/0°–45° |
| Shoulder | Internal/external rotation | 0°–80°/0°–30° | 0°–80°/0°–30° |
| Elbow | Flexion/extension | 0°–135° | 0°–105° |
| Elbow | Internal/external rotation | 0°–45°/0°–45° | 0°–45°/0°–45° |
| Wrist | Flexion/extension | 0°–75°/0°–75° | 0°–45°/0°–45° |

## 3. Kinematic Analysis

Because the object of robot's service is the injured limb, the injured limb wears on the robot and moves together under its traction to achieve rehabilitation training. It is a basis of motion control and execution of rehabilitation training. In order to enable the rehabilitation robot to perform more efficient motion control in the process of rehabilitation training, the movement between the robot's end and each joint can be coordinated by establishing the spatial pose relationship between the robot's motion components and the end-effector. The movement variation of each joint of the wearable rehabilitation robot can be appropriately changed, and the movement between the end of the wearable rehabilitation robot and each joint can be adjusted to achieve the expected rehabilitation training requirements.

### 3.1. Forward Kinematics

The wearable upper limb rehabilitation robot is a typical human-machine cooperation system. The robot is consistent with the movement of the affected limb of the human body. Therefore, to accurately obtain the motion curve of the affected limb, a forward kinematic analysis is required for the wearable upper limb rehabilitation robot. To ensure that the designed wearable upper limb rehabilitation robot has good applicability and practical applications, the D-H parameter model of the wearable upper limb rehabilitation robot based on the D-H coordinate system method needs to

set related parameters, which including describing the connecting rod, which used to describe the geometric characteristic parameters of connecting rods, the connection parameter relationship between two connecting rods and the parameters that define the relationship between connecting rods. Lastly, the parameters can be brought into the correlation transformation matrix to get the corresponding results by setting the parameters.

In order to obtain the relationship between the rotation and translation among adjacent members of the wearable upper limb rehabilitation robot and its terminal pose, a reference coordinate system should be assigned to each joint [36]. In this paper, according to kinematics theory, Denavit–Hartenberg (D-H) coordinate system method is adopted to establish the D-H parameter model of the upper limb rehabilitation robot ontology (see Figure 6), as shown in Figure 7. The geometric size of each connecting rod of the rehabilitation robot can be described by four parameters. $a_i$ and $\alpha_i$ are used to describe the geometric features of the connecting rod itself. The numerical values are determined by the distance and included angle between the axes $z_{i-1}$ and $z_i$ The other parameters offset $d_i$ and joint angle $\theta_i$ represent the connection relationship between the two connecting rods, and the values are determined by the distance and included angle between the axes $x_{i-1}$ and $x_i$. The parameters of the rehabilitation robot D-H model are listed in Table 2.

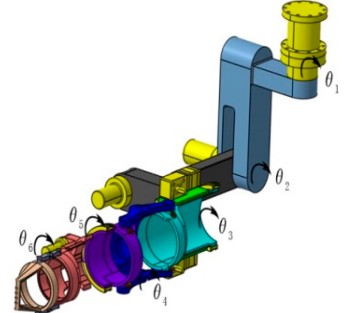

**Figure 6.** 3D rehabilitation robot.

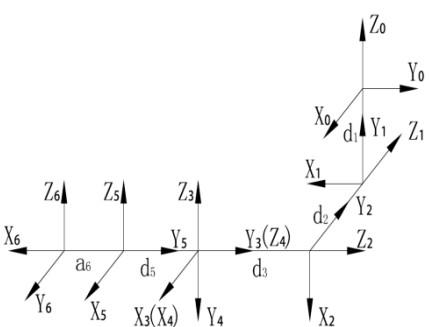

**Figure 7.** Denavit–Hartenberg (D-H) parameter model of robot.

**Table 2.** D-H parameters of upper limb rehabilitation robot.

| Joint $i$ | Length of Bar $a_i$ | Knob Angle $\alpha_i$ | Offset $d_i$ | Joint Angle $\theta_i$ | Angle Range |
|:---:|:---:|:---:|:---:|:---:|:---:|
| 1 | 0 | 90° | 235 mm | $\theta_1$ (−90°) | −90°–30° |
| 2 | 0 | 90° | 20 mm | $\theta_2$ (−90°) | −90°–45° |
| 3 | 0 | 90° | 420 mm | $\theta_3$ (−90°) | −80°–45° |
| 4 | 0 | 90° | 0 | $\theta_4$ (0°) | 0°–135° |
| 5 | 0 | 90° | 265 mm | $\theta_5$ (0°) | −45°–45° |
| 6 | 83 mm | 0° | 0 | $\theta_6$ (−90°) | −45°–45° |

According to the spatial coordinate system established for each joint of the wearable upper limb rehabilitation robot, forward kinematics analysis is carried out, i.e., the coordinate transformation from

coordinate system $\{O_{i-1}\}$ to coordinate system $\{O_i\}$. Based on the robotics theory, the vector described in coordinate system $\{O_i\}$ is mapped to coordinate system $\{O_{i-1}\}$ by coordinate transformation:

$$
\begin{aligned}
^{i-1}T_i \quad &= A_i = \mathrm{Rot}(Z, \theta_i) \times \mathrm{Trans}(0, 0, d_i) \times \mathrm{Trans}(a_i, 0, 0) \times \mathrm{Rot}(X, a_i) \\
&= \begin{bmatrix} c\theta_i & -s\theta_i & 0 & 0 \\ s\theta_i & c\theta_i & 0 & 0 \\ 0 & 0 & 1 & 0 \\ 0 & 0 & 0 & 1 \end{bmatrix} \begin{bmatrix} 1 & 0 & 0 & a_i \\ 0 & 1 & 0 & 0 \\ 0 & 0 & 1 & d_i \\ 0 & 0 & 0 & 1 \end{bmatrix} \begin{bmatrix} 1 & 0 & 0 & 0 \\ 0 & c\theta_i & -s\alpha_i & 0 \\ 0 & s\alpha_i & c\alpha_i & 0 \\ 0 & 0 & 0 & 1 \end{bmatrix} \\
&= \begin{bmatrix} c\theta_i & -s\theta_i c\alpha_i & s\theta_i s\alpha_i & a_i c\theta_i \\ s\theta_i & c\theta_i c\alpha_i & -c\theta_i s\alpha_i & a_i s\theta_i \\ 0 & s\alpha_i & c\alpha_i & d_i \\ 0 & 0 & 0 & 1 \end{bmatrix}
\end{aligned}
\tag{1}
$$

where c means a cosine function and s represents sine function.

For a wearable upper limb rehabilitation robot, when the coordinate system of each link is determined, the parameters of each link can be obtained. According to Formula (1), it can be known that the pose matrix $A_i$ between the two rods is:

$$
A_1 = \mathrm{Rot}(Z, \theta_1) \times \mathrm{Trans}(0, 0, d_1) \times \mathrm{Trans}(a_1, 0, 0) \times \mathrm{Rot}(X, \alpha_1) = \begin{bmatrix} c\theta_1 & 0 & s\theta_1 & 0 \\ s\theta_1 & 0 & -c\theta_1 & 0 \\ 0 & 1 & 0 & 235 \\ 0 & 0 & 0 & 1 \end{bmatrix}
$$

$$
A_2 = \mathrm{Rot}(Z, \theta_2) \times \mathrm{Trans}(0, 0, d_2) \times \mathrm{Trans}(a_2, 0, 0) \times \mathrm{Rot}(X, \alpha_2) = \begin{bmatrix} c\theta_2 & 0 & s\theta_2 & 0 \\ s\theta_2 & 0 & -c\theta_2 & 0 \\ 0 & 1 & 0 & 20 \\ 0 & 0 & 0 & 1 \end{bmatrix}
$$

$$
A_3 = \mathrm{Rot}(Z, \theta_3) \times \mathrm{Trans}(0, 0, d_3) \times \mathrm{Trans}(a_3, 0, 0) \times \mathrm{Rot}(X, \alpha_3) = \begin{bmatrix} c\theta_3 & 0 & s\theta_3 & 0 \\ s\theta_3 & 0 & -c\theta_3 & 0 \\ 0 & 1 & 0 & 420 \\ 0 & 0 & 0 & 1 \end{bmatrix}
$$

$$
A_4 = \mathrm{Rot}(Z, \theta_4) \times \mathrm{Trans}(0, 0, d_4) \times \mathrm{Trans}(a_4, 0, 0) \times \mathrm{Rot}(X, \alpha_4) = \begin{bmatrix} c\theta_4 & 0 & s\theta_4 & 0 \\ s\theta_4 & 0 & -c\theta_4 & 0 \\ 0 & 1 & 0 & 0 \\ 0 & 0 & 0 & 1 \end{bmatrix}
$$

$$
A_5 = \mathrm{Rot}(Z, \theta_5) \times \mathrm{Trans}(0, 0, d_5) \times \mathrm{Trans}(a_5, 0, 0) \times \mathrm{Rot}(X, \alpha_5) = \begin{bmatrix} c\theta_5 & 0 & s\theta_5 & 0 \\ s\theta_5 & 0 & -c\theta_5 & 0 \\ 0 & 1 & 0 & 265 \\ 0 & 0 & 0 & 1 \end{bmatrix}
$$

$$
A_6 = \mathrm{Rot}(Z, \theta_6) \times \mathrm{Trans}(0, 0, d_6) \times \mathrm{Trans}(a_6, 0, 0) \times \mathrm{Rot}(X, \alpha_6) = \begin{bmatrix} c\theta_6 & -s\theta_6 & 0 & 83c\theta_6 \\ s\theta_6 & c\theta_6 & 0 & 83s\theta_6 \\ 0 & 0 & 1 & 0 \\ 0 & 0 & 0 & 1 \end{bmatrix}
$$

Therefore, the matrix transformation calculation formula can be obtained by the robot's end handle posture relative to the robot's base coordinate system:

$$
{}^0_6T = A_1A_2A_3A_4A_5A_6 =
\begin{bmatrix}
r_{11} & r_{12} & r_{13} & P_x \\
r_{21} & r_{22} & r_{23} & P_y \\
r_{31} & r_{32} & r_{33} & P_z \\
0 & 0 & 0 & 1
\end{bmatrix}
\tag{2}
$$

where $\begin{bmatrix} r_{11} & r_{12} & r_{13} \\ r_{21} & r_{22} & r_{23} \\ r_{31} & r_{32} & r_{33} \end{bmatrix}$ is the direction vector of the terminal, and, $\begin{bmatrix} p_x & p_y & p_z \end{bmatrix}^T$ is the position vector of the terminal.

To obtain the solution of positive kinematics, the matrices are multiplied as follows:

$$r_{11} = s\theta_6(s\theta_4(s\theta_1s\theta_3+c\theta_1c\theta_2c\theta_3) - c\theta_1c\theta_4s\theta_2) - c\theta_6(s\theta_5(c\theta_3s\theta_1-c\theta_1c\theta_2s\theta_3)$$
$$-c\theta_5(c\theta_4(s\theta_1s\theta_3+c\theta_1c\theta_2c\theta_3) + c\theta_1s\theta_2s\theta_4))$$

$$r_{12} = s\theta_6(s\theta_5(c\theta_3s\theta_3-c\theta_1c\theta_2s\theta_3) - c\theta_5(c\theta_4(s\theta_1s\theta_3+c\theta_1c\theta_2c\theta_3) + c\theta_1si\theta_2s\theta_4))$$
$$+c\theta_6(s\theta_4(s\theta_1s\theta_3+c\theta_1c\theta_2c\theta_3) - c\theta_1c\theta_4s\theta_2)$$

$$r_{13} = c\theta_5(c\theta_3s\theta_3-c\theta_1c\theta_2si\theta_3) + s\theta_5(c\theta_4(s\theta_1s\theta_3+c\theta_1c\theta_2c\theta_3) + c\theta_1s\theta_2s\theta_4)$$

$$r_{21} = c\theta_6(s\theta_6(c\theta_1c\theta_3+c\theta_2s\theta_1s\theta_3) - c\theta_5(c\theta_4(c\theta_1s\theta_3-c\theta_2c\theta_3s\theta_1) - s\theta_1s\theta_2s\theta_4))$$
$$-s\theta_6(si\theta_4(c\theta_1s\theta_3) - c\theta_2c\theta_3s\theta_1) + c\theta_4s\theta_1s\theta_2)$$

$$r_{22} = -s\theta_6(s\theta_5(c\theta_1c\theta_3+c\theta_2s\theta_1s\theta_3) - c\theta_5(c\theta_4(c\theta_1s\theta_3-c\theta_2c\theta_3s\theta_1) - s\theta_1s\theta_2s\theta_4))$$
$$-c\theta_6(s\theta_4(c\theta_1s\theta_3-c\theta_2c\theta_3s\theta_1) + c\theta_4s\theta_1s\theta_2)$$

$$r_{23} = -c\theta_5(c\theta_1c\theta_3+c\theta_2s\theta_1s\theta_3) - s\theta_5(c\theta_4(c\theta_1s\theta_3-c\theta_2c\theta_3s\theta_1) - s\theta_1s\theta_2s\theta_4)$$

$$r_{31} = s\theta_6(c\theta_2c\theta_4+c\theta_3s\theta_2s\theta_4) - c\theta_6(c\theta_5(c\theta_2s\theta_4-c\theta_3c\theta_4s\theta_2) - s\theta_2s\theta_3s\theta_5)$$

$$r_{32} = c\theta_6(c\theta_2c\theta_4+c\theta_3s\theta_2s\theta_4) + s\theta_6(c\theta_5(c\theta_2s\theta_4-c\theta_3c\theta_4s\theta_2) - s\theta_2s\theta_3s\theta_5)$$

$$r_{33} = -s\theta_5(c\theta_2s\theta_4-c\theta_3c\theta_4s\theta_2) - c\theta_5s\theta_2s\theta_3$$

$$P_x = 20s\theta_1 + 265s\theta_4(s\theta_1s\theta_3 + c\theta_1c\theta_2c\theta_3)$$
$$-83c\theta_6(s\theta_5(c\theta_3s\theta_1 - c\theta_1c\theta_2s\theta_3) - c\theta_5(c\theta_4(s\theta_1s\theta_3 + c\theta_1c\theta_2c\theta_3) + c\theta_1s\theta_2s\theta_4))$$
$$+83s\theta_6(s\theta_4(s\theta_1s\theta_3 + c\theta_1c\theta_2c\theta_3) - c\theta_1c\theta_4s\theta_2) + 420c\theta_1s\theta_2 - 265c\theta_1c\theta_4s\theta_2$$

$$P_y = 420s\theta_1s\theta_2-20c\theta_1-265s\theta_4(c\theta_1s\theta_3-c\theta_2c\theta_3s\theta_1) + 83c\theta_6(s\theta_5(c\theta_1c\theta_3+c\theta_2s\theta_1s\theta_3)$$
$$-c\theta_5(c\theta_4(c\theta_1s\theta_3-c\theta_2c\theta_3s\theta_1) - s\theta_1s\theta_2s\theta_4)) - 83s\theta_6(s\theta_4(c\theta_1s\theta_3-c\theta_2c\theta_3s\theta_1) + c\theta_4s\theta_1s\theta_2) - 265c\theta_4s\theta_1s\theta_2$$

$$P_z = 83s\theta_6(c\theta_2c\theta_4+c\theta_3s\theta_2s\theta_4) - 420c\theta_2-83c\theta_6(c\theta_5(c\theta_2s\theta_4-c\theta_3c\theta_4s\theta_2) - s\theta_2s\theta_3s\theta_5)$$
$$+265c\theta_2c\theta_4+265c\theta_3s\theta_2s\theta_4+235$$

### 3.2. Workspace Analysis

The workspace of the rehabilitation robot is the set of points that the hand reference points can reach in space during the operation of the rehabilitation robot [37]. It is a key element to estimate the robot rehabilitation training. Moreover, it provides a key point for evaluating the rationality of machine design. In this paper, the calculation is the set of the end can reach the target point when the rehabilitation robot is given all pose, i.e., the full workspace. The Monte Carlo method is exploited to analyze the workspace of the rehabilitation robot, being a kind of numerical calculation method guided by probability and statistics theory. Taking the end of the wearable upper limb rehabilitation robot as the reference point and combining the position vector in the forward kinematics equation, the set of all random points that generate the reference point constitutes the workspace of the wearable upper limb rehabilitation robot. In the specified range, the angle value of each joint is generated by random sampling method, and then the angle value is substituted into the positive solution calculation

program to obtain the corresponding end position point. After 100,000 simulation computations, which can be obtained the terminal spot cloud.

The Programming and Robotics Toolbox of MATLAB software is utilized to adjust the random points. When the sample size is 100,000, the simulation diagram of the wearable upper limb rehabilitation robot's workspace is obtained, as shown in Figure 8.

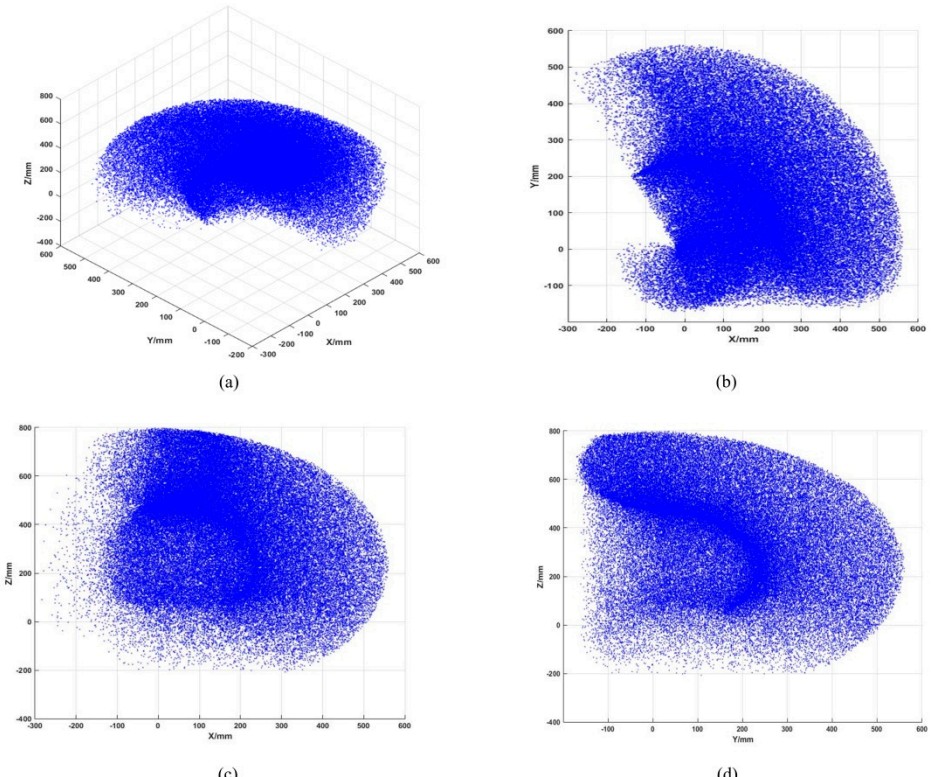

**Figure 8.** A 3D workspace simulation of the robot: (**a**) 3D spatial diagram; (**b**) XOY view; (**c**) XOZ view; (**d**) YOZ view.

Geometric construction can be calculated more easily to construct the boundary of the maximum range. Due to the linear relationship between the components, the terminal pose corresponding to the joint angles cannot be predicted in extreme cases. However, this design method is to simulate the possible arrival space, and give the relevant probability point cloud.

As can be seen from Figure 8, the workspace is a sector in XOZ view, and it is approximate in XOY view and YOZ view, which is roughly an ellipse removing part of both ends of the long axis. According to Figure 8 and kinematic analysis, it concludes that the working radius of the robot's workspace on the X-axis and Y-axis is 795 mm, and the working radius on the Z-axis is 552 mm. According to ergonomics, the average length of the medium human arm in China is 742 mm [38], indicating that the limit position of the wearable upper limb rehabilitation robot is very close to the dynamic limit position of the human upper limb. Therefore, the wearable upper limb rehabilitation robot designed to meet the needs of upper limb rehabilitation. Meanwhile, due to the limitation of mechanical structure and its size, a cavity appears near the base of the wearable upper limb rehabilitation robot. The size of the cavity is relevant to the size of the base.

### 3.3. Inverse Kinematics

Wearable upper limb rehabilitation robots often assist patients with hemiplegia to complete some tasks in daily life, such as eating, holding things, and touching the head, to achieve the purpose of restoring upper limb motor function. If the task is given, the end position is reached by the wearable

upper limb rehabilitation robot at this time. In order to determine the parameters of each joint angle under the known position of the end effector, the wearable upper limb rehabilitation robot is solved of inverse kinematics.

The inverse kinematics solution mainly includes two kinds of numerical solutions and closed solutions. When the numerical method is used, the specific value of the joint variable can be obtained by using a recursive algorithm. The result of the numerical calculation can only be used. The independent variable cannot be given at random and the calculated value can be obtained [39]. On the other hand, the closed solution method can be deduced according to the formula, and the dependent variable can be obtained by giving any independent variable. Because the closed solution is more accurate and faster than the numerical method, and it is easy to distinguish all possible solutions, the wearable upper limb rehabilitation robot designed in this paper satisfies the Pieper criterion in robot kinematics, therefore, this paper uses the closed solution to solve, which provide a basis for subsequent structural optimization and motion control.

Given the parameter values of D-H $\theta_i$ can be obtained by inversely solving the following matrices:

$$^0_6T = ^0_1T(\theta_1)^1_2T(\theta_2)^2_3T(\theta_3)^3_4T(\theta_4)^4_5T(\theta_5)^5_6T(\theta_6) \tag{3}$$

Then,

$$\theta_1 = \tan^{-1}\left(\frac{-P_y}{P_x}\right) + k\pi \tag{4}$$

$$\theta_2 = \tan^{-1}\left(\frac{164\sin\varphi - 29\cos\varphi - b}{a + 164\cos\varphi + 29\sin\varphi}\right) + k\pi \tag{5}$$

$$\theta_3 = \tan^{-1}\left(\frac{x-y}{z-u}\right) + k\pi \tag{6}$$

$$\theta_4 = \tan^{-1}\left(\frac{s_4 s_5}{c_4 s_5}\right) + k\pi \tag{7}$$

$$\theta_5 = \tan^{-1}\left(\frac{\frac{r_{33}}{c_5} + c_2 s_3 + c_3 s_2}{(s_2 s_3 - c_2 c_3)c_4}\right) + k\pi \tag{8}$$

$$\theta_6 = \tan^{-1}\left(\frac{r_{32}v + r_{31}w}{-r_{32}w + r_{31}v}\right) + k\pi \tag{9}$$

where $k$ means the integer value, and the value of $k$ must be within the range of the value of $\theta_i$ at the joint angle; $a = \frac{P_y}{-5s_1} - 110$, $b = \frac{P_z}{5}$; $\varphi = \sin^{-1}m - \gamma + k\pi$, $m = \frac{1163-a^2-b^2}{\sqrt{(328a+58b)^2+(58a-328b)^2}}$, $\gamma = \tan^{-1}\left(\frac{328a+58b}{58a-328b}\right)$; $x = (170c_2 - a)(164s_2 - 29c_2)$; $y = (170s_2 + b)(164c_2 + 29s_2)$; $z = (170s_2 + b)(29c_2 - 164s_2)$; $u = (170c_2 - a)(164c_2 + 29s_2)$, $v = c_2 s_3 s_5 + c_3 s_2 s_5 - c_2 c_3 c_4 c_5 + c_4 c_5 s_2 s_3$.

The parameters of a certain number of position points which can be considered as the end of the wearable upper limb rehabilitation robot are substituted into the above formula to obtain the corresponding theoretical rotation angles of each joint. It is consistent with the rotation angle of each joint in the 3D model. Thereby, it verifies the correctness of the inverse kinematics solution of the wearable upper limb rehabilitation robot.

Since there are multiple inverse solutions in the process of solving inverse kinematics, this article introduces three criteria for selecting solutions for robots to avoid singular solutions.

(1) Robot motion range requirements. During the structural design of the wearable upper limb rehabilitation robot, the motion range of each joint is designed according to the requirements of the upper limb motion range of the human body and to avoid mutual interference between mechanisms. Therefore, the joints are required to meet the requirements of the joint motion range during the movement, that is, the following inequality is satisfied:

$$\theta_i^{\min} \le \theta_i \le \theta_i^{\max} \; i = 1, 2, \cdots, 6 \tag{10}$$

where $\theta_i^{\min}$ is the minimum limit angle of joint motion, and $\theta_i^{\max}$ is the maximum limit angle of joint motion.

(2) Criterion of motion continuity. The application environment of the wearable upper limb rehabilitation robot is to assist functional rehabilitation training for patients with motor dysfunction. It requires the robot to continuously run smoothly and smoothly during the movement, no joint angle motion mutation, and to avoid the movement of the end effector of the robot. Therefore, in the actual processing, the threshold value m is introduced to provide constraints and restrictions, that is, to check the absolute value of the difference between the next solution and the current joint angle value. If the value is too large, the joint is abruptly changed, then another solution is selected.

$$\Delta\theta = |\theta_n - \theta_{cur}| \le m \tag{11}$$

where $\theta_n$ is the next position angle of joint motion, and $\theta_{cur}$ is the current position angle of joint motion.

(3) The principle of minimum end pose error. During the movement of the wearable upper limb rehabilitation robot, the position reached by the end effector and the target pose will inevitably have errors. Therefore, by comparing the final end pose transformation matrix, the end position and attitude error can be compared to achieve the goal of choosing the solution with the smallest error.

During the process of the selection for the inverse solution of the wearable upper limb rehabilitation robot, the constrained conditions will also conflict with each other. At this time, the priority of the constraints should be set according to the actual situation, so as to obtain the optimal solution.

## 4. Simulation Analysis of the Rehabilitation Robot Based on Virtual Prototype

### 4.1. Kinematic Simulation and Optimization Design

In the structure of wearable upper limb rehabilitation robot, each joint is directly driven by motor. In order to avoid excessive rigid impact on the affected limb, which causes secondary injury, ADAMS virtual prototype simulation software is exploited to simulate and analyze the motion stability of wearable upper limb rehabilitation robot, and optimize the structure accordingly. The torque of each joint and the center of mass at the end of the rehabilitation robot were selected as the measurement objects, and the whole motion time was set at 5 s. The driving motor rotated at a constant speed to study the motion state of the shoulder joint and elbow joint. The simulation results are shown in Figures 9 and 10.

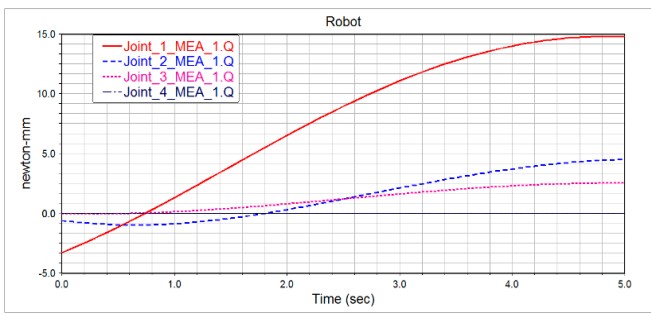

**Figure 9.** Torque transformation curves of each joint.

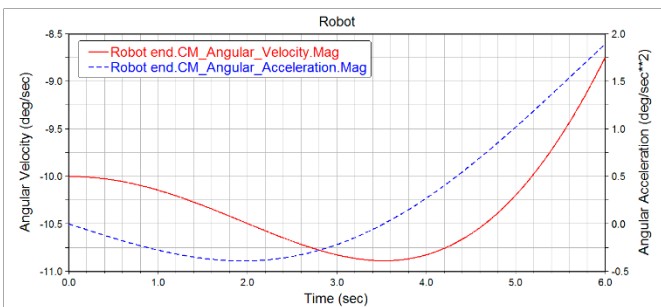

**Figure 10.** End angular velocity/angular acceleration curve.

Specifically, Figure 9 shows the curve of joint torque changing with time when each joint is moving. The initial state of the model is arm abduction to the state of maximum angle, and then adduction to the position of maximum angle. Joint1, Joint2 and Joint3 are the torque variation curves when the shoulder joint moves around three axes of three degrees of freedom simultaneously, and Joint4 is the torque variation curve when the elbow joint moves at one degree of freedom. As can be observed, except Joint1 of shoulder freedom, the torque of each joint fluctuates little at the start, then gradually flattens to a certain value, and the motion curve is always smooth. It indicates that during the uniform motion of each joint, the torque fluctuation of the joint is small and there is no big impact on the arm. Among them, the torque of the elbow joint retains almost unchanged, indicating that the torque required by the wrist joint is very small during the movement. This infers that during the uniform motion of each joint, the torque fluctuation of the joint is small and there is no obviously impact on the arm.

Figure 10 plots the curves of angular velocity and angular acceleration changing with time when the end is moving. The initial state of the model is arm abduction to the state of maximum angle, and then adduction to the position of maximum angle. It can be seen from Figure 10 that the torque of each joint fluctuates greatly at startup, and then the motion curve is smooth along with the movement of each joint. This indicates that the end torque fluctuates less during the uniform motion of each joint, and has no significantly impact on the hand.

*4.2. Dynamic Simulation Analysis*

In this paper, the dynamic analysis uses the three-dimensional model established in Section 2 into Adams (Automatic dynamic analysis of mechanical systems) software, and then set the parameters of the imported model, and add driving to each joint of the wearable upper limb rehabilitation robot. In order to select the motor and ensure the output torque to satisfy the needs of rehabilitation, the dynamic simulation of the wearable upper limb rehabilitation robot was carried out. Considering the differences between patients with joint bearing capacity, for most people to bear upon initial value, limb resistance to wearable upper limb rehabilitation robot is set to: 100 N in the horizontal movement direction, 200 N in the vertical direction, the end points are in the hand close to the elbow, shoulder rotation direction of 4 Nm torque, stress points in the forearm center of mass. The initial position of the simulation is the wearable upper limb rehabilitation robot arm extending horizontally to the front, and the simulation results are shown in Figure 11. The solid line in the diagram represents the torque, while the dotted line indicates the rotation angle of the robot.

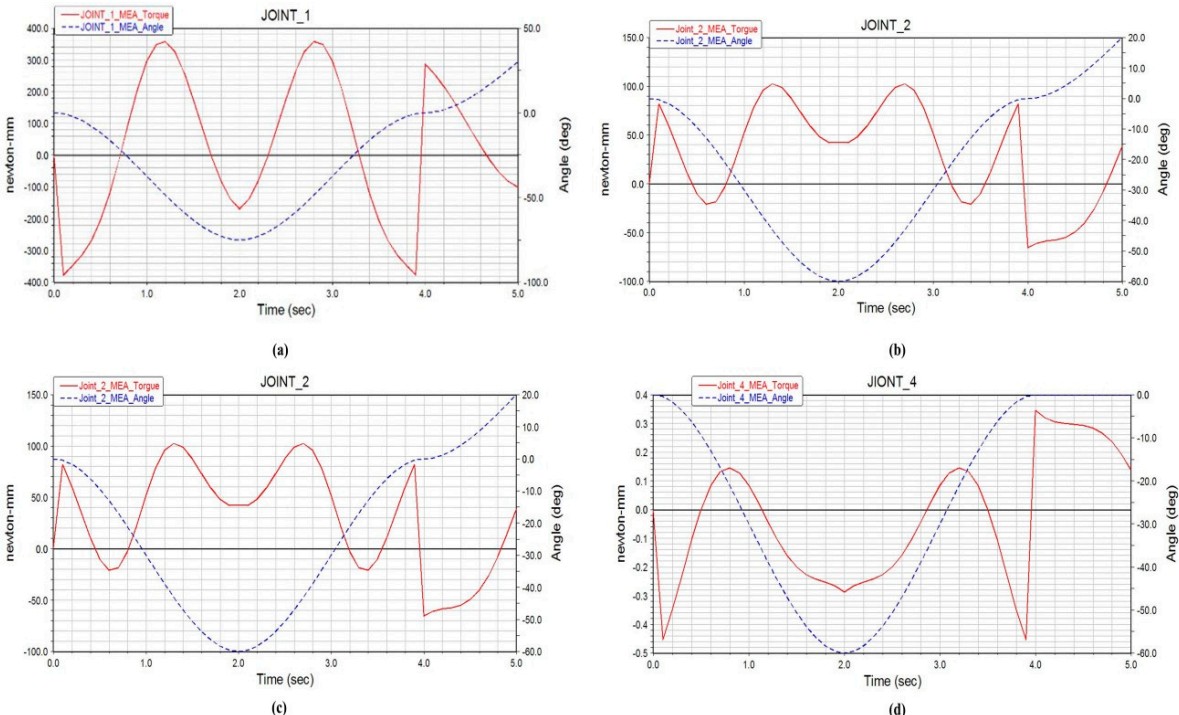

**Figure 11.** Torques and angles of joints. (**a**) The rotation angle and the output torque of the drive motor when the shoulder joint moves in the vertical direction at different moments; (**b**) the relationship between the angle of rotation and the torque on the worm gear when the shoulder moves horizontally; (**c**) the relationship between the rotation angle of the shoulder joint and the output torque of the motor; (**d**) relationship between elbow flexion and extension angle and driving torque.

As shown in Figure 11, the dotted line is the joint rotation angle change curve, the variable is set to $\theta$, the solid line is the joint torque change curve $T$, and the variable is set to $T = G \cdot L(\theta(t))$. From the torque formula $L$, where is the distance from the center of mass of the robot arm to the axis of the shoulder joint.

$$
\begin{aligned}
{}^{i-1}T_i &= A_i = \text{Rot}(Z, \theta_i) \times \text{Trans}(0, 0, d_i) \times \text{Trans}(a_i, 0, 0) \times \text{Rot}(X, a_i) \\
&= \begin{bmatrix}
c\theta_i & -s\theta_i c\alpha_i & s\theta_i s\alpha_i & a_i c\theta_i \\
s\theta_i & c\theta_i c\alpha_i & -c\theta_i s\alpha_i & a_i s\theta_i \\
0 & s\alpha_i & c\alpha_i & d_i \\
0 & 0 & 0 & 1
\end{bmatrix}
\end{aligned}
$$

${}^{i-1}T_i$ is a transformation matrix between adjacent arm joints based on the D-H parameter model.

$$
L = \sum \{T\_i \, [\theta(t)]\}
$$

The maximum angle is $\theta = 120°$ in the initial position, and returns to the original position $\theta = -280°$, and then gos to the maximum angle $\theta = 120°$.

More specifically, Figure 11a shows the relationship between the rotation angle and the output torque of the drive motor when the shoulder joint moves in the vertical direction at different moments. As can be seen from the Figure 11a, when the arm abducts to the maximum angle, the torque required is larger, and the mechanism is furthest from the static balance position. Figure 11b shows the relationship between the angle of rotation and the torque on the worm gear when the shoulder moves horizontally. The wearable upper limb rehabilitation robot first swings up to the lowest point, then returns to the initial point, and finally swings down to the maximum angle. The maximum torque of movement in the horizontal direction occurs near the position from the top swing to the highest point, where

the required continuous torque time is relatively large, indicating that the motor continues to work. Figure 11c shows the relationship between the rotation angle of the shoulder joint and the output torque of the motor. The forearm rotates 600 in, 600 out and back to the starting position. The motor delivers torque through the gears and arc-shaped rack, providing a stable torque output for the elbow. Figure 11d shows the relationship between elbow flexion and extension angle and driving torque. With the increase of the angle, the load of the hand end deviates from the static balance position and the torque increases. As can be seen from Figure 11, through gear rack and other variable gear, the output torque of the drive motor is between a few Nm to dozens of Nm. The vertical direction of the shoulder joint is larger, and the output torque of the direct drive method is generally in the tens of *N*.m to hundreds of Nm, which can be seen that the output performance requirements of the motor are greatly reduced. According to the simulation results, the DC stepper servo motor with planetary reducer is selected for the driving motor, which guarantees the accuracy of motion while ensuring the power.

As the virtual prototype is an ideal model, and the simplified driving function is adopted for joint rotation. During joint rotation, the PID controller in the actual machine is not adopted when the rotation direction is changed, which leads to the local region mutation, which isn't the focus of this paper, therefore, it is omitted here.

## 5. Experimental Setup and Results

The wearable upper limb rehabilitation robot designed in this paper adopts the upper computer-controlled robot to carry out continuous passive rehabilitation training for the affected limb, and adopts two rehabilitation training methods: independent movement of single joint and combined functional movement of multiple joints. The block diagram of the overall control system is shown in Figure 12. The lower computer of the control circuit takes the motion control card and data acquisition card as the core. The information acquisition card is used to complete data collection of 3D gyroscope, motion acceleration and other sensors. The upper computer can select the rehabilitation training mode and set the motion parameters of each joint, and then transmit the related parameters to the motion control card. The data collected by each functional sensor will be transmitted to the data acquisition card. After calculation, the data acquisition card converts the collected data into motion parameters, and then passes the motion control card to the motor rotation information, which is output to each motor driver respectively.

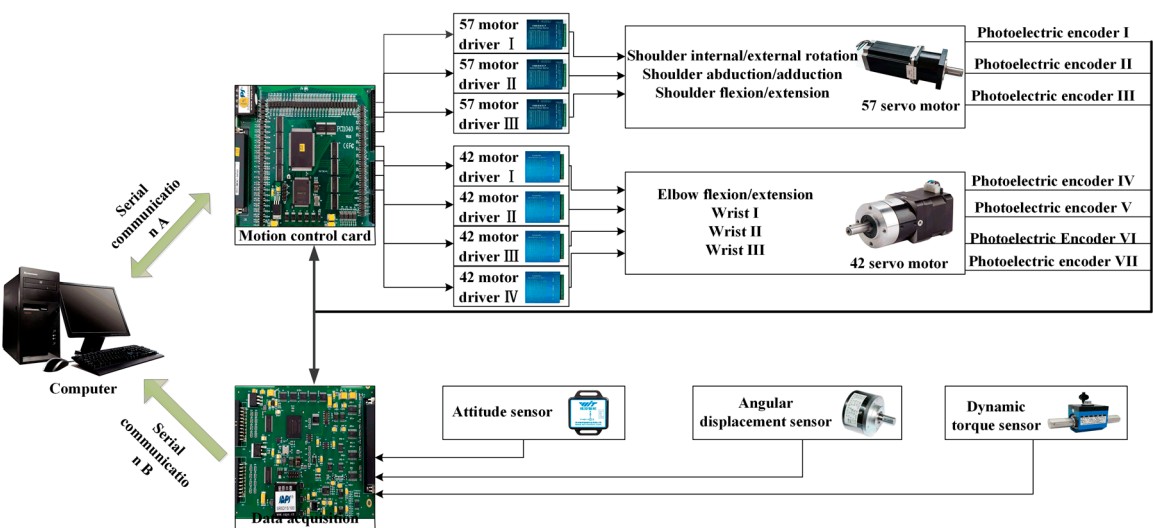

**Figure 12.** Block diagram of rehabilitation robot control system.

To realize the precise control of wearable upper limb rehabilitation robots, and reduce the motor torque error between the actual output torque and the theory, and let patients achieve the best effect of rehabilitation, the torque demand larger shoulder joint is designed in this paper. In addition, the drive motor of bending/stretching exercise and elbow flexion/stretch exercise drive motor installed on the dynamic torque sensor based on the basis of the Section 4 results, that is to say, wearable upper limbs rehabilitation robot dynamics analysis results of output torque. The dynamic torque sensor transmits the information to the computer through the data acquisition card. The difference between the theoretical torque and the measured torque is input into the PID controller and then the torque of the motor is adjusted in real time to achieve the optimal control.

When designing a control system, it is essential to ensure the stability of the system and the safety of patients. Some control methods and neural network technique have been paid more attention to designing controller for a nonlinear system [40,41]. In order to prevent excessive output torque of the motor from causing secondary damage to the affected limb, a current detection circuit is adopted to feed the output current of the driver into the motion control card in real time. Once the detected current exceeds the allowable value, the motor will stop immediately. At the same time, the one-key emergency stop function is set so that the system can be stopped immediately when the patient is uncomfortable. Furthermore, gyroscope and linear velocity sensor are used to detect the real-time position, real-time speed and real-time acceleration of the motor, so as to avoid the adverse effect caused by excessive joint rotation and excessive speed on the rehabilitation training.

Figure 13 depicts the completed wearable upper limb rehabilitation robot prototype system and rehabilitation training. In the experiment, the time of each degree of freedom of the wearable upper limb rehabilitation robot was set to 25 s, and the range of motion was from the initial position to the extreme position. The test sampling period was 1 s. The experimental results are shown in Figure 14. Specifically, the experiment curves of horizontal, vertical, rotation, and elbow flexion and extension of the upper limb are plotted respectively. The solid red line in the figure is the simulation result, and the blue dotted line is the test result. It can be observed, the actual angle and simulation angle of each joint of the wearable upper limb rehabilitation robot during the movement have certain deviations. It is mainly due to the error caused by the friction of the mechanical structure itself, and the subject wearing the wearable upper limb rehabilitation robot will also be difficult to avoid rehabilitation. The wearable upper limb rehabilitation robot applies a certain impedance, and one can see that the elbow joint flexion and extension error is relatively large, because this part is driven by rope and the wire rope needs to bear a large load, which increases the error caused by the elastic contraction of the wire rope itself, but these errors do not affect the training characteristics of wearable upper limb rehabilitation robot. The test results of the prototype test are consistent with the simulation results, revealing that the wearable upper limb rehabilitation robot meets the expected design requirements and can complete the motion functions required for passive rehabilitation training of the upper limbs.

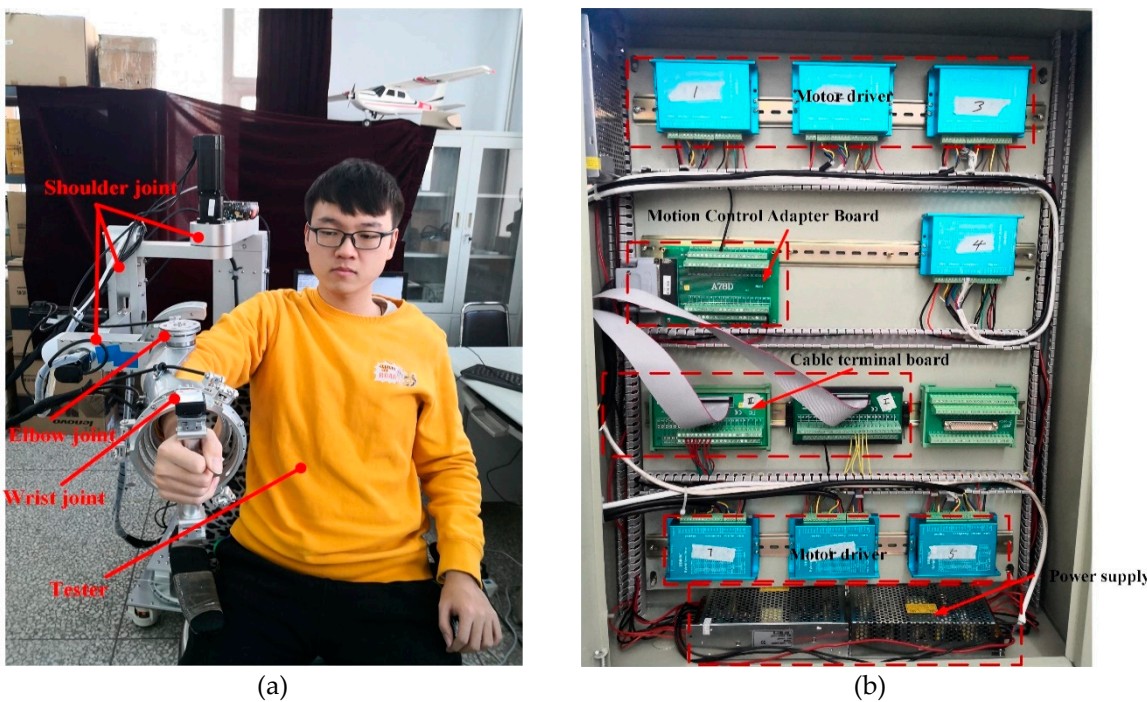

**Figure 13.** Test and control system diagram of rehabilitation robot. (**a**) Tester wearing prototype test; (**b**) physical map of control system.

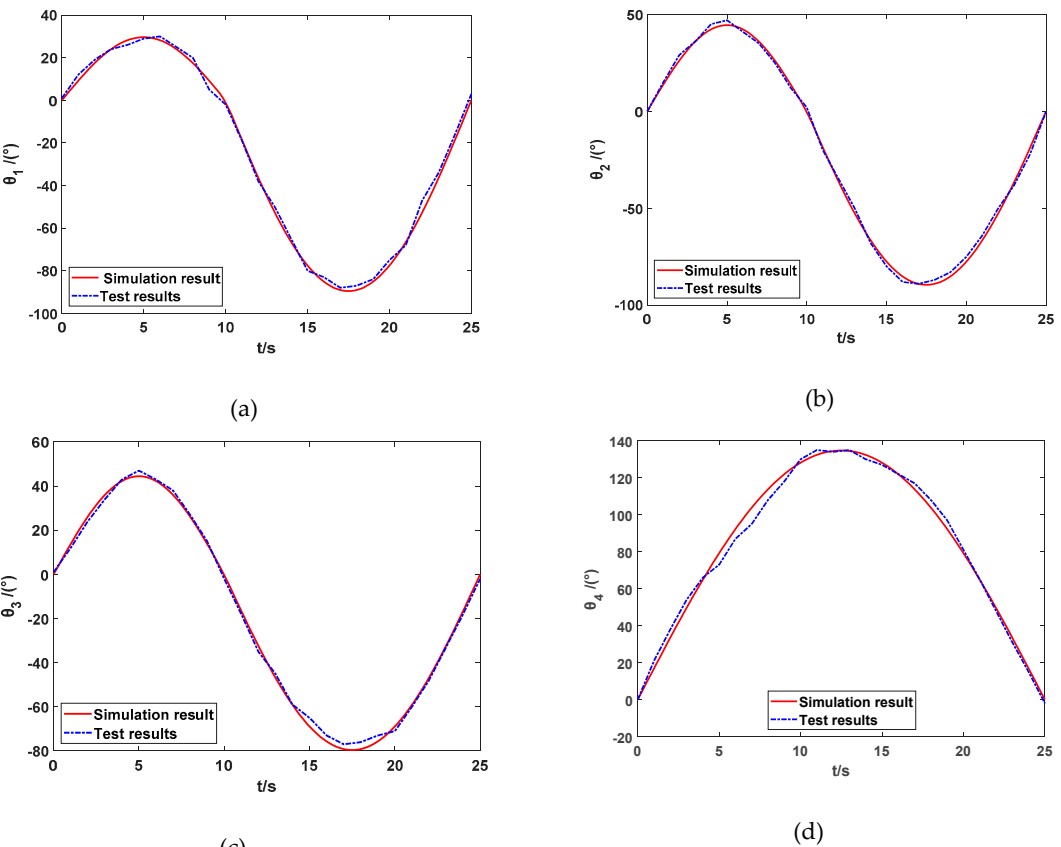

**Figure 14.** Tracking results of wearable upper limb rehabilitation robot. (**a**) Shoulder flexion/extension; (**b**) shoulder abduction/adduction; (**c**) shoulder internal/external rotation; (**d**) elbow flexion/extension.

## 6. Conclusions

Combined rehabilitation medicine and bionics, a 6-degree-of-freedom upper limb rehabilitation robot was proposed, analyzed and investigated via using ropes as the mainstay and driving by the "rope + toothed belt". Combined with the human upper limb muscle anatomy characteristic and relevant parameters, the mechanical mechanism of the wearable upper limb rehabilitation robot was designed during the process of training movement. The workspace and kinematics analyses of wearable upper limb rehabilitation robot were completed and analyzed, which laid the foundation for the control scheme method of rehabilitation robot. The kinematics analysis of the shoulder joint and joint motion was solely carried out in the horizontal and vertical directions, which resulted in the relationship between the elbow joint flexion and extension angle and the driving torque. It also provided a basis for the selection of the motor. The design of the wearable upper limb rehabilitation robot control system was completed to ensure the safety of the affected limb during rehabilitation training movement. The rehabilitation training experiments showed that the results were consistent with the numerical simulation results, which could satisfy the requirements of rehabilitation training. In addition, the movement transition between wearable upper limb rehabilitation robot and upper limb was smooth, which might not lead secondary injury to the injured upper limb.

In future work, the structure optimization will be conducted to make the wearable upper limb rehabilitation robot more compact and anthropomorphic. Furthermore, the control algorithm of the wearable upper limb rehabilitation robot will be extensively investigated to enhance the control precision and interaction ability of the robot system. Eventually, the advancement of the wearable upper limb rehabilitation robot will in turn provide impetus for the development of rehabilitation medical equipment in the context of upper limb rehabilitation training.

**Author Contributions:** In this work, Z.P., and T.W. conceived and designed the experiments; J.Y. gave some constructive suggestions; Z.S. performed the experiments; Z.P., and S.L. analyzed the data; Z.W. guided the writing of the article and made some modifications; and Z.P. wrote the paper. All authors have read and agreed to the published version of the manuscript.

**Funding:** This work was supported in part by the National Natural Science Foundation of China under Grant 51875047 and Grant 61873304, in part by the China Postdoctoral Science Foundation funded project under Grant 2019T120240 and 2018M641784, and in part by the Foundation of Jilin Province Science and Technology under Grant 20170307012YY.

**Acknowledgments:** The authors are grateful to the anonymous reviewers and the Editor for their valuable comments and suggestions on improving this paper.

**Conflicts of Interest:** The authors declare no conflict of interest.

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
