# Peer review of "Design and Analysis of a Wearable Upper Limb Rehabilitation Robot with Characteristics of Tension Mechanism"

_applsci, doi:10.3390/app10062101_

Round 1

Reviewer 1 Report

The authors presented a work on a wearable upper limb robot.

General comment:
althought it is clear that the authors put a lot of effort in developing the system and present the results, I do not think that this manuscript is ready for a publication, since it resembles more a technical report of a project without any discussion of the results. The objective appears to be the "only" design of the rehabilitation robot, a task that has been deeply studied (for instance, see works by Vitiello et al. or Frisoli et al.). The authors do not provide any specific application or implication of such a device, limiting the dissertation to a list of standard activities to the detriment of the novelty/ies that is/are not clear.

Other comments:

  • 76-77: "the main problem is that the motion axis of each motion pair in exoskeleton robot must match with the rotation center of each joint of human body, which poses great challenges to the design of exoskeleton rehabilitation robot". I agree but there is no clue on how the authors faced this important aspect.
  • 90 - 115: this part of the text is more a discussion/conclusion part. I would rather suggest to improve the introduction with more relevant papers (to be discussed later on). 
  • Section 2: add references to the main statements related to literature
  • Figure 1: please improve the caption using, possibly, numbering to the panels
  • Figure 2: in the caption, the title and panel A have the same description. Please improve all the caption. In panel b, please move the label "geared motor" that overlaps the figure.
  • 172: "important role", please elaborate more
  • Figure 3 & 4 and Figures 5 & 6 can be merged using panels
  • Figure 7: please use "frame" to label the main structure, avoiding "reinforced rib", "base" "upright" and "cantilever". You mention the shoulder adaptive mechanism, but it is not described in the text.
  • Table 1: add reference to column 3
  • Figures 5 & 6 can be merged using panels
  • Section 3 is a standard approach and it is not scientifically relevant. I suggest to move it to supplementary
  • Section 4 can be put at the end of Section 2 after Table
  • 337: please add reference
  • 362-379: please use numerical references to describe the plots
  • 386,419,420: "Nm" (no italic) instead of "N.m"

Minor issues:

  • Use "s" instead of "sec" throughout the document
  • Check grammar and style throughout the document (e.g., Monte carol at lines 325-326)

Author Response

Dear Editors and Reviewers,

On behalf of all the contributing authors, I would like to express our sincere appreciation for your letter and reviewers’ constructive comments concerning our article entitled “Design and analysis of a wearable upper limb rehabilitation robot with characteristics of tension mechanism” (Manuscript No.: applsci-730801).

According to these comments, we have made extensive modifications to our manuscript and supplemented extra simulations to make our results convincing. Point-by-point responses to the comments of associate editor and reviewers are listed below with the corresponding revisions highlighted in blue colour in the revised manuscript.

Response to Reviewer 1 Comments

Point 1: Although it is clear that the authors put a lot of effort in developing the system and present the results, I do not think that this manuscript is ready for a publication, since it resembles more a technical report of a project without any discussion of the results. The objective appears to be the "only" design of the rehabilitation robot, a task that has been deeply studied (for instance, see works by Vitiello et al. or Frisoli et al.). The authors do not provide any specific application or implication of such a device, limiting the dissertation to a list of standard activities to the detriment of the novelty/ies that is/are not clear.

 Response 1: The authors sincerely thank the reviewer for raising such a comment. We have made correction according to the reviewer’s comments.

To satisfy the rehabilitation needs of patients with limb disorders, a wearable upper limb rehabilitation robot is designed and developed in this article, which is mainly a device for mid-term semi-active rehabilitation training and post-active rehabilitation training for stroke patients. Owing to understanding the disadvantages of traditional rehabilitation training and the performances of rehabilitation robots, combined with the human upper limb muscle anatomy characteristic and relevant parameters, we determine the arm movement of each joint angle range from all the bones and joints of upper limb movement characteristics, this paper proposes a design scheme of tensegrity structure wearable upper limbs rehabilitation robot. The wearable upper limb rehabilitation robot is utilized to the exercise rehabilitation treatment of hemiplegic limb to maintain the range of motion of the limb, prevent the muscle atrophy of the limb, enhance the muscle strength of the limb, and promote the recovery of the limb function. Therefore, it can provide an effective rehabilitation equipment for patients with hemiplegia of upper limb caused by stroke.

In this paper, due to the study of anatomy, motion mechanism and motion range of human upper limb, the motion angle range of each joint is determined for human arm, and the mechanical mechanism on each degree of freedom is designed for wearable upper limb rehabilitation robot. First, to establish the spatial pose relationship between each motion component and the end-effector of the wearable upper limb rehabilitation robot, the motion model is established with D-H parameter method and the motion space is analyzed for wearable upper limb rehabilitation robot. The kinematics analysis is used to analyze the motion of the wearable upper limb rehabilitation robot. Second, to verify whether the wearable upper limb rehabilitation robot can realize the auxiliary upper limb functional rehabilitation training, the working space is analyzed for the wearable upper limb rehabilitation robot. Third, to analyze the output torque of wearable upper limb rehabilitation robot, the dynamic simulation of the robot is carried out. Last, the control system of wearable upper limb rehabilitation robot is designed, which obtained the tracking results of robot rehabilitation training. It further verifies that the rationality of the design of wearable upper limb rehabilitation robot.

The main contributions of this paper are summarized as follows:

(1) Owing to the anatomy theory, motion mechanism and range of human upper limbs, a novel wearable upper limb rehabilitation robot with tension mechanism is firstly designed, investigated and analyzed for upper limb injured patients based on flexible transmission during rehabilitation training process. A cable-driven modular parallel joints are innovatively designed for elbow/wrist and a shoulder joint driven by a toothed belt. All the cable-driven motors are rear-mounted to achieve long-distance transmission and reduce the drive inertia of the end joints. The gear belt is exploited to drive the joints of a wearable upper limb rehabilitation robot, which realizing high precision meshing. The design approach of the wearable upper limb rehabilitation robot facilitates the rehabilitation training of the joint, effectively reduces the volume, mass and inertia of the actuators, and achieves the lightweight design of the overall structure.

(2) Besides, this paper proposes a flexibly parallel mechanism of humanoid wrist driven by rope and supported through compression spring. The fixed base and moving platform of the wearable upper limb rehabilitation robot are connected by three ropes and a conical compression spring. The springs are designed by simulating the human wrist and support the mobile platform to complete the wrist movement, while the ropes are constructed via simulating the wrist muscles to control the wearable upper limb rehabilitation robot. In this paper, the design approach will contribute to the further study of parallel mechanisms with flexible joints. The results will play an important role in reappearing the movement of human wrist and promote the development of rehabilitation robot and rope drive technology.

(3) The kinematics and workspace of the wearable upper limb rehabilitation robot are verified and analyzed based on D-H method and Monte Carlo method. It demonstrates that the wearable upper limb rehabilitation robot can satisfy the requirements of rehabilitation training through kinematics/dynamics analysis and rehabilitation training experiments. Therefore, it also further verifies that the feasibility and effectiveness of the design method, which provides a valuable idea for improving rehabilitation robot mechanism.

 Point 2: 76-77: "the main problem is that the motion axis of each motion pair in exoskeleton robot must match with the rotation center of each joint of human body, which poses great challenges to the design of exoskeleton rehabilitation robot". I agree but there is no clue on how the authors faced this important aspect.

Response 2: The authors sincerely thank the reviewer for the time, effort and the frank recognition given to the manuscript. We have made correction according to the reviewer’s comments.

In the design of exoskeleton prostheses, the matching of mechanical joint motion axis and human joint motion axis is very important. The exoskeleton produces unexpected forces at the patient's joint under mismatched condition, which not only causes joint pain and injury to the patient, but also limits the movement space of the patient's limbs, and reduces the effect of rehabilitation training. Therefore, the axis of each pair of motion is matched with the rotation center of each joint of the human body as far as possible in the design of exoskeleton rehabilitation apparatus. The motion of each joint of exoskeleton rehabilitation device is realized mainly by rotating or moving the pair, and good results have been obtained [23-24]。

Point 3: 90 - 115: this part of the text is more a discussion/conclusion part. I would rather suggest to improve the introduction with more relevant papers (to be discussed later on).

Response 3: The authors sincerely thank the reviewer for the time, effort and the frank recognition given to the manuscript. We have made correction according to the reviewer’s comments.

Point 4: Section 2: add references to the main statements related to literature.

Response 4: The authors sincerely thank the reviewer for such a helpful suggestion. We have made correction according to the reviewer’s comments.

Point 5: Figure 1: please improve the caption using, possibly, numbering to the panels.

Response 5: The authors sincerely thank the reviewer for such a helpful suggestion. We have made correction according to the reviewer’s comments.

Figure 1. Shoulder joint freedom of motion. (a) Flexion/extension; (b) Abduction/adduction; (c) Internal rotation/external rotation.

Point 6: Figure 2: in the caption, the title and panel A have the same description. Please improve all the caption. In panel b, please move the label "geared motor" that overlaps the figure.

Response 6: The authors sincerely thank the reviewer for such a helpful suggestion. We have made correction according to the reviewer’s comments. The details can be seen as follows:

Figure 2. The structure diagram of shoulder joint. (a) 3D model of shoulder joint;(b) 3D model of shoulder joint rotation degree of freedom.

Point 7: 172: "important role", please elaborate more.

Response 7: The authors sincerely thank the reviewer for such a helpful suggestion. We have made correction according to the reviewer’s comments. The details can be seen as follows:

"Important role" mainly expresses that elbow joint mainly completes such as eating, holding things, touching the head and so on in people's daily life. If the elbow joint movement is restricted, it will be greatly restricted, and other joints, which will also have a greater impact for the patient's daily life. Therefore, the elbow joint plays an important role in the upper limb joint.

Point 8: Figure 3 & 4 and Figures 5 & 6 can be merged using panelsï¼›Figure 7: please use "frame" to label the main structure, avoiding "reinforced rib", "base" "upright" and "cantilever". You mention the shoulder adaptive mechanism, but it is not described in the text.

Response 8: The authors sincerely thank the reviewer for such a helpful suggestion. We have made correction according to the reviewer’s comments. The details can be seen as follows:

Figure 3. Elbow joint structure diagram. (a) Elbow joint freedom of motion; (b) 3D model of elbow joint.

Figure 4. Wrist joint structure diagram. (a) Wrist joint freedom of motion; (b) 3D model of wrist joint.

Figure 7. 3D model of wearable upper limb rehabilitation robot.

The "adaptation" mentioned in this paper refers to the adjustment of the height adjustment mechanism of the seat, which adapts to the different lengths of the human upper limbs and the different body shapes of the human body, and is suitable to different treatment environments and patient body shape. Furthermore, it ensures that the injured limb on the sagittal plane during the rehabilitation training.

Point 9: Table 1: add reference to column 3.

Response 9: The authors sincerely thank the reviewer for such a helpful suggestion. We have made correction according to the reviewer’s comments.

Point 10: Section 3 is a standard approach and it is not scientifically relevant. I suggest to move it to supplementary.

Response 10: The authors sincerely thank the reviewer for such a helpful suggestion. We have made correction according to the reviewer’s comments. The details can be seen as follows:

Section 3 mainly describes the kinematics analysis of the wearable upper limb rehabilitation robot. Because the object of robot's service is the injured limb, the injured limb wears on the robot and moves together under its traction to achieve rehabilitation training. It is a basis of motion control and execution of rehabilitation training. In order to enable the rehabilitation robot to perform more efficient motion control in the process of rehabilitation training, the movement between the robot's end and each joint can be coordinated by establishing the spatial pose relationship between the robot's motion components and the end-effector. The movement variation of each joint of the wearable rehabilitation robot can be appropriately changed, and the movement between the end of the wearable rehabilitation robot and each joint can be adjusted to achieve the expected rehabilitation training requirements.Therefore, the content of this section is reserved.

Point 11: Section 4 can be put at the end of Section 2 after Table.

Response 11: The authors sincerely thank the reviewer for such a helpful suggestion. We have made correction according to the reviewer’s comments.

Point 12: 337: please add reference.

Response 12: The authors sincerely thank the reviewer for such a helpful suggestion. We have made correction according to the reviewer’s comments.

Point 13: 362-379: please use numerical references to describe the plots.

Response 13: The authors sincerely thank the reviewer for such a helpful suggestion. We have made correction according to the reviewer’s comments. The details can be seen as follows:

As shown in Figure 11, the dotted line is the joint rotation angle change curve, the variable is set to , the solid line is the joint torque change curve , and the variable is set to . From the torque formula , where is the distance from the centre of mass of the robot arm to the axis of the shoulder joint.

a is a transformation matrix between adjacent arm joints based on the D-H parameter model。

The maximum angle is  in the initial position, and returns to the original position , and then gos to the maximum angle .

Point 14: 386,419,420: "Nm" (no italic) instead of "N.m".

Response 14: The authors sincerely thank the reviewer for such a helpful suggestion. We have made correction according to the reviewer’s comments.

Point 15: Use "s" instead of "sec" throughout the documentï¼›Check grammar and style throughout the document (e.g., Monte carol at lines 325-326).

Response 15: The authors sincerely thank the reviewer for his/her frank recognition given to the manuscript, in addition to the time and effort spent in reviewing our manuscript.

Finally, we would like to say thanks again sincerely to the editor and anonymous reviewers for their time and effort spent in handling our manuscript, as well as providing us many constructive comments for improving very much the presentation and quality of this manuscript. According to reviewers’ suggestions, we tried our best to improve the manuscript and made some changes in the manuscript. These changes will not influence the content and framework of the paper. We appreciate for Editors’ warm work earnestly and hope that the correction will meet with approval. Once again, thank you very much for your comments and suggestions.

Reviewer 2 Report

This paper deals with the design of wearable upper limb rehabilitation robot with 6 dof. The kinematics as well as dynamics simulation have been performed. The corresponding workspace is analysed and represented.  

The contribution of the work is not clearly defined and the final results still questionable. The authors are requested to more justify their choices, hypotheses as well as the mathematical formulation

There is some incomprehension from the abstract that authors should explain in detail:

  • The perspective of bionics: what authors means by this and no work linked to biomechanical approach or study has been addressed in the paper  
  • Rules of minimum driving moment: this appears as a constraint applied to an optimisation problem but there no optimal solution or optimisation function to be defined.  

The paper presents an interesting work but some items below still questionable and request further developments

Section 2.1, works linked to biomechanical studies are missing. I would suggest including more literature reviews, at least to cover some of those widely known works, such as:  

- Laribi M A, Decatoire A, Carbone G, Pisla D, Zeghloul S. Identification of upper limb motion specifications via visual tracking for robot assisted exercising. International Conference on Robotics in Alpe-Adria-Danube Region, Patras, Greece, 2018.

- Gates DH, Walters LS, Cowley J, Wilken JM, Resnik L. Range of Motion Requirements for Upper-Limb Activities of Daily Living. Am J Occup Ther 2016;70(1):7001350010p1-7001350010p10. doi:10.5014/ajot.2016.015487 .

- Major K A, Major Z Z, Carbone G, Pisla A, Vaida C, Gherman B, Pisla D L. Ranges of motion as basis for robot-assisted post-stroke rehabilitation. Human & Veterinary Medicine, 2016, 8, 192–196.

Section 2.1, what authors means by “passive gear”

Section 2.1, in line 160 authors introduced safety. How safety of patients is ensured and can be verified.

The role of pulley needs to be more detailed in the proposed design, at figure 2.

The relation between the actuator and the driving wheel at the elbow joint is missing.

Section 2.3, the daily activities cited at line 204 should be more explained and analysed. On what kind of motion the proposed system is focused?

Section 2.3, A conical compression spring is used in the design of the wrist joint. Authors are invited to determine the location of the Remote centre of motion and justify the use of conical instead of cylindrical.

Section 3.1, the description of the DH parameters (lines 266-270) as well as equation 1 are not needed.    

Section 3.1, what authors means by “positive solution” in line 285.

Section 3.2, this section is focused on the inverse kinematic model but no discussion on the singular configurations is addressed. The multiple solutions of the IKM need to be discussed.

Section 4, line 325 : “Monte Carlo” instead of “Monte Carol”

Section 4, the boundary of the workspace can be defined using geometric construction. The use of the Monte Carlo method is not justified. The obtained workspace given on Fig 10 is not useful.

Section 5.2, there are some discontinuities in the torque curves especially in joints 1,2 and 4. How authors explain and cope with these issues. Authors should be careful with the selection of the actuator and detail the adopted procedure.

Section 5.2, the inertial parameters included in the dynamic model are missing and authors are invited to detail clearly the model.

Section 6, a dynamic torque sensor is used as mentioned on the block diagram. Authors are invited to detail the number of used torque sensors and way to use in control scheme.

Section 6, the cartesian trajectory of the end-effector is necessary in addition to a comparison between the test and simulation results.

I would like to encourage the authors to find all answers and revise their work.

Author Response

Dear Editors and Reviewers,

On behalf of all the contributing authors, I would like to express our sincere appreciation for your letter and reviewers’ constructive comments concerning our article entitled “Design and analysis of a wearable upper limb rehabilitation robot with characteristics of tension mechanism” (Manuscript No.: applsci-730801).

According to these comments, we have made extensive modifications to our manuscript and supplemented extra simulations to make our results convincing. Point-by-point responses to the comments of associate editor and reviewers are listed below with the corresponding revisions highlighted in blue colour in the revised manuscript.

Response to Reviewer 2 Comments

Point 1: This paper deals with the design of wearable upper limb rehabilitation robot with 6 dof. The kinematics as well as dynamics simulation have been performed. The corresponding workspace is analysed and represented. 

The contribution of the work is not clearly defined and the final results still questionable. The authors are requested to more justify their choices, hypotheses as well as the mathematical formulation

Response 1: The authors sincerely thank the reviewer for raising such a comment. We have made correction according to the reviewer’s comments.

In this paper, due to the study of anatomy, motion mechanism and motion range of human upper limb, the motion angle range of each joint is determined for human arm, and the mechanical mechanism on each degree of freedom is designed for wearable upper limb rehabilitation robot. First, to establish the spatial pose relationship between each motion component and the end-effector of the wearable upper limb rehabilitation robot, the motion model is established with D-H parameter method and the motion space is analyzed for wearable upper limb rehabilitation robot. The kinematics analysis is used to analyze the motion of the wearable upper limb rehabilitation robot. Second, to verify whether the wearable upper limb rehabilitation robot can realize the auxiliary upper limb functional rehabilitation training, the working space is analyzed for the wearable upper limb rehabilitation robot. Third, to analyze the output torque of wearable upper limb rehabilitation robot, the dynamic simulation of the robot is carried out. Last, the control system of wearable upper limb rehabilitation robot is designed, which obtained the tracking results of robot rehabilitation training. It further verifies that the rationality of the design of wearable upper limb rehabilitation robot.

Since the service object of the robot is the injured limb, the injured limb is worn on the robot and moves together under its traction to realize the rehabilitation training. Kinematic analysis is the basis of motion control and execution in rehabilitation training. To make the performances of rehabilitation robot more efficiently in the process of rehabilitation training of motion control, the space position relations is established through robot motion artifacts and the end executor, which causes the robot end and coordinate with each other between each joint movement, and appropriately changes each joint movement variation. Therefore, the wearable rehabilitation robots adjusts the end-effector with the movement of each joint, which realize the expected requirements of rehabilitation training.

Point 2: There is some incomprehension from the abstract that authors should explain in detail: The perspective of bionics: what authors means by this and no work linked to biomechanical approach or study has been addressed in the paper.

Response 2: The authors sincerely thank the reviewer for the time, effort and the frank recognition given to the manuscript. We have made correction according to the reviewer’s comments.

Owing to study of the human upper extremity anatomy, movement mechanism, and the range of motion, it can determine the range of motion angles of the human arm joints, and design the shoulder joint, elbow joint, and wrist joint separately under the principle of ensuring the minimum driving torque. Then, the kinematics, workspace and dynamics analysis of each structure are performed. Finally, the control system of the rehabilitation robot is designed. The experimental results show that the structure is convenient to wear on the human body, and the robot's freedom of movement matches well with the freedom of movement of the human body. It can effectively support and traction the front and rear arms of the affected limb, and accurately transmit the applied traction force to the upper limb of the joints. The rationality of the wearable upper limb rehabilitation robot design is verified, which can help patients achieve rehabilitation training and provide an effective rehabilitation equipment for patients with hemiplegia caused by stroke.

Point 3: Rules of minimum driving moment: this appears as a constraint applied to an optimisation problem but there no optimal solution or optimisation function to be defined.

Response 3: The authors sincerely thank the reviewer for the time, effort and the frank recognition given to the manuscript. We have made correction according to the reviewer’s comments.

The principle of minimum driving torque described in this article refers to selecting a smaller driving motor and a reducer on the premise that the output torque of the selected driving motor can meet the rehabilitation training task of the wearable upper limb rehabilitation robot. When we reduce the structure size, it makes the wearable upper limb rehabilitation robot look more beautiful and more affinity.

Point 4: Section 2.1, works linked to biomechanical studies are missing. I would suggest including more literature reviews, at least to cover some of those widely known works, such as:

Laribi M A, Decatoire A, Carbone G, Pisla D, Zeghloul S. Identification of upper limb motion specifications via visual tracking for robot assisted exercising. International Conference on Robotics in Alpe-Adria-Danube Region, Patras, Greece, 2018.

Gates DH, Walters LS, Cowley J, Wilken JM, Resnik L. Range of Motion Requirements for Upper-Limb Activities of Daily Living. Am J Occup Ther 2016;70(1):7001350010p1-7001350010p10. doi:10.5014/ajot.2016.015487 .

Major K A, Major Z Z, Carbone G, Pisla A, Vaida C, Gherman B, Pisla D L. Ranges of motion as basis for robot-assisted post-stroke rehabilitation. Human & Veterinary Medicine, 2016, 8, 192–196.

Response 4: he authors sincerely thank the reviewer for such a helpful suggestion. We have made correction according to the reviewer’s comments.

Point 5: Section 2.1, what authors means by “passive gear”.

Response 5: The authors sincerely thank the reviewer for the time, effort and suggestions spent in reviewing our manuscript.

Since the position of the shoulder joint internal/external rotation mechanism is far away from the base, the quality of the mechanism will have a great impact on the control during the movement. After repeated experiments, it is determined that the arc joint rack is used to achieve the shoulder joint internal/external rotation movement. Because the curved rack needs to cooperate with the pulley set, the driving gear is not allowed to directly mesh with the curved rack in the assembly space. To this end, we have added a passive gear between the active gear and the curved rack, which can reduce the output torque of the motor reduces the quality of the mechanism and the structure is more compact and beautiful.

Point 6: Section 2.1, in line 160 authors introduced safety. How safety of patients is ensured and can be verified.

Response 6: The authors sincerely thank the reviewer for such a helpful suggestion. We have made correction according to the reviewer’s comments. The details can be seen as follows:

The transmission mechanism of the shoulder joint internal/external rotation mechanism is active gear-passive gear-arc rack, where both ends of the arc racks are provided with shoulders. The rack gear meshes. Once it exceeds the rehabilitation range, the passive gear will be blocked by the shoulder and cannot continue to move, which ensuring the safety of the patient and avoiding secondary injuries to the patient.

Point 7: The role of pulley needs to be more detailed in the proposed design, at figure 2.

Response 7: The authors sincerely thank the reviewer for his/her frank recognition given to the manuscript, in addition to the time and effort spent in reviewing our manuscript.

In order to ensure a certain motion accuracy, two sets of pulley blocks are installed on both sides of the arc-shaped rack. The function is to restrict the movement of the arc-shaped guide rail along the direction of the arc, and play the role of limit.

Point 8: The relation between the actuator and the driving wheel at the elbow joint is missing.

Response 8: The authors sincerely thank the reviewer for the time, effort and suggestions spent in reviewing our manuscript. The details can be seen as follows:

The driven part of the elbow joint movement mechanism is mounted on the base, and the two-way driven plate of the motor transmits the power to the two-way wire plate of the elbow through the rope, thus, it completes the elbow flexion/extension motion.

Figure 3. The structure diagram of elbow joint. (a) The elbow joint freedom of motion; (b) 3D model of elbow joint.

Point 9: Section 2.3, the daily activities cited at line 204 should be more explained and analysed. On what kind of motion the proposed system is focused?

Response 9: The authors sincerely thank the reviewer for the time, effort and suggestions spent in reviewing our manuscript. The details can be seen as follows:

In this paper, people mainly complete such actions as eating, taking things and touching their heads in the daily life. The wrist joint not only has a high frequency of motion, but also is the part of the upper limb that bears the largest load in the process of supporting, pushing and pulling. The design of wearable upper limbs rehabilitation robot mainly for medium-term and semi-active rehabilitation training in patients with cerebral apoplexy and late active rehabilitation training device, furthermore, the wrist of patients should have certain activity. During rehabilitation training, patients need to hold the end adjusting grip of the wearable upper limb rehabilitation robot, and the upper limb follows the robot to do corresponding rehabilitation training.

Point 10: Section 2.3, A conical compression spring is used in the design of the wrist joint. Authors are invited to determine the location of the Remote centre of motion and justify the use of conical instead of cylindrical.

Response 10: The authors sincerely thank the reviewer for the time, effort and suggestions spent in reviewing our manuscript. The details can be seen as follows:

As shown in Figure 4 (b), a flexible parallel mechanism is proposed to simulate human wrist with rope drive. The wrist adopts the hand-wrist-forearm connection. The front and rear sections of the wrist are connected by a tapered compression spring, which is used to simulate the motion of a wrist joint. There are three sets of rope mechanism around, each set of rope mechanism is separated by 120o to simulate the wrist muscles, which complete the drive and control of the wrist. In addition, control mechanism is equipped with a power source to be placed in the base part. The flexible parallel mechanism takes the human wrist as the bionic object, where the fixed ring is equivalent to the radius and ulnar complex, and the moving ring is equivalent to the metacarpal bone. The driving rope and spring represent the muscles and ligaments around the wrist respectively, which providing kinetic energy and support for the motion of the radial and middle wrist joints. The parallel mechanism uses three servo motors to drive three ropes, which realize the wrist flexion/extension and ulnar/radial movement of the robot. Besides, it can not only reduce the flexible degree of freedom, but also enhance the stability of the mechanism, and make the mechanism satisfy the motion amplitude of the wrist under different angles when the mechanism is in retraction and abduction, flexing and stretching. Therefore, the mechanism can achieve the wrist joint adduction and abduction, flexion straight action. The flexible wrist joint driven by rope is mainly composed of three parts: adjustable grip, flexible parallel mechanism and forearm fixation.

Figure 4. The structure diagram of wrist joint. (a) Wrist joint freedom of motion; (b) 3D model of wrist joint.

Owing to load and deformation are nonlinear, and comparing with cylindrical helical spring, the conical helical spring has a greater stability and prevents resonance phenomenon, the application is more and more widely. Specially, if the load does not make the spring coil contact, the relationship of the load and deformation is linear, and if the load continues to increase, then the spring contact from a large ring, the relationship of load and deformation is nonlinear.

Point 11: Section 3.1, the description of the DH parameters (lines 266-270) as well as equation 1 are not needed.

Response 11: The authors sincerely thank the reviewer for the time, effort and suggestions spent in reviewing our manuscript. The details can be seen as follows:

The wearable upper limb rehabilitation robot is a typical human-machine cooperation system. The robot is consistent with the movement of the affected limb of the human body. Therefore, to accurately obtain the motion curve of the affected limb, a forward kinematic analysis is required for the wearable upper limb rehabilitation robot. To ensure that the designed wearable upper limb rehabilitation robot has good applicability and practical applications, the D-H parameter model of the wearable upper limb rehabilitation robot based on the D-H coordinate system method needs to set related parameters, which including describing the connecting rod, which used to describe the geometric characteristic parameters of connecting rods, the connection parameter relationship between two connecting rods and the parameters that define the relationship between connecting rods. At last, the parameters can be brought into the correlation transformation matrix to get the corresponding results by setting the parameters.

Point 12: Section 3.1, what authors means by “positive solution” in line 285.

Response 12: The authors sincerely thank the reviewer for such a helpful suggestion. We have made correction according to the reviewer’s comments. The details can be seen as follows:

                                                                                         ï¼ˆ1)

where c means a cosine function and s represents sine function.

For a wearable upper limb rehabilitation robot, when the coordinate system of each link is determined, the parameters of each link can be obtained. According to formula (1), it can be known that the pose matrix  between the two rods is:

Therefore, the matrix transformation calculation formula can be obtained by the robot's end handle posture relative to the robot's base coordinate system:

                                                                                               ï¼ˆ2)

where  is the direction vector of the terminal, and,  is the position vector of the terminal.

To obtain the solution of positive kinematics, the matrices are multiplied as follows:

Point 13: Section 3.2, this section is focused on the inverse kinematic model but no discussion on the singular configurations is addressed. The multiple solutions of the IKM need to be discussed.

.

Response 13: The authors sincerely thank the reviewer for such a helpful suggestion. We have made correction according to the reviewer’s comments. The details can be seen as follows:

Wearable upper limb rehabilitation robots often assist patients with hemiplegia to complete some tasks in daily life, such as eating, holding things, and touching the head, to achieve the purpose of restoring upper limb motor function. If the task is given, the end position is reached by the wearable upper limb rehabilitation robot at this time. In order to determine the parameters of each joint angle under the known position of the end effector, the wearable upper limb rehabilitation robot is solved of inverse kinematics.

The inverse kinematics solution mainly includes two kinds of numerical solutions and closed solutions. When the numerical method is used, the specific value of the joint variable can be obtained by using a recursive algorithm. The result of the numerical calculation can only be used. The independent variable cannot be given at random and the calculated value can be obtained. On the other hand, the closed solution method can be deduced according to the formula, and the dependent variable can be obtained by giving any independent variable. Because the closed solution is more accurate and faster than the numerical method, and it is easy to distinguish all possible solutions, the wearable upper limb rehabilitation robot designed in this paper satisfies the Pieper criterion in robot kinematics, therefore, this paper uses the closed solution to solve, which provide a basis for subsequent structural optimization and motion control.

Since there are multiple inverse solutions in the process of solving inverse kinematics, this article introduces three criteria for selecting solutions for robots to avoid singular solutions.

(1) Robot motion range requirements. During the structural design of the wearable upper limb rehabilitation robot, the motion range of each joint is designed according to the requirements of the upper limb motion range of the human body and to avoid mutual interference between mechanisms. Therefore, the joints are required to meet the requirements of the joint motion range during the movement, that is, the following inequality is satisfied:

where  is the minimum limit angle of joint motion, and  is the maximum limit angle of joint motion.

 (2) Criterion of motion continuity. The application environment of the wearable upper limb rehabilitation robot is to assist functional rehabilitation training for patients with motor dysfunction. It requires the robot to continuously run smoothly and smoothly during the movement, no joint angle motion mutation, and to avoid the movement of the end effector of the robot. Therefore, in the actual processing, the threshold value m is introduced to provide constraints and restrictions, that is, to check the absolute value of the difference between the next solution and the current joint angle value. If the value is too large, the joint is abruptly changed, then another solution is selected.

where  is the next position angle of joint motion, and  is the current position angle of joint motion.

 (3) The principle of minimum end pose error. During the movement of the wearable upper limb rehabilitation robot, the position reached by the end effector and the target pose will inevitably have errors. Therefore, by comparing the final end pose transformation matrix, the end position and attitude error can be compared to achieve the goal of choosing the solution with the smallest error.

During the process of the selection for the inverse solution of the wearable upper limb rehabilitation robot, the constrained conditions will also conflict with each other. At this time, the priority of the constraints should be set according to the actual situation, so as to obtain the optimal solution.

Point 14: Section 4, line 325 : “Monte Carlo” instead of “Monte Carol”.

Response 14: The authors sincerely thank the reviewer for the time, effort and suggestions spent in reviewing our manuscript.

Point 15: Section 4, the boundary of the workspace can be defined using geometric construction. The use of the Monte Carlo method is not justified. The obtained workspace given on Fig 10 is not useful.

Response 15: The authors sincerely thank the reviewer for such a helpful suggestion. We have made correction according to the reviewer’s comments. The details can be seen as follows:

In the specified range, the angle value of each joint is generated by random sampling method, and then the angle value is substituted into the positive solution calculation program to obtain the corresponding end position point. After 100,000 simulation computations, which can be obtained the terminal spot cloud.

Geometric construction can be calculated more easily to construct the boundary of the maximum range. Due to the linear relationship between the components, the terminal pose corresponding to the joint angles cannot be predicted in extreme cases. However, this design method is to simulate the possible arrival space, and give the relevant probability point cloud, furthermore, the program design is relatively simple.

Point 16: Section 5.2, there are some discontinuities in the torque curves especially in joints 1,2 and 4. How authors explain and cope with these issues. Authors should be careful with the selection of the actuator and detail the adopted procedure.

Response 16: The authors sincerely thank the reviewer for the time, effort and suggestions spent in reviewing our manuscript. We have made correction according to the reviewer’s comments.

As the virtual prototype is an ideal model, and the simplified driving function is adopted for joint rotation. During joint rotation, the PID controller in the actual machine is not adopted when the rotation direction is changed, which leads to the local region mutation, which is the focus of this paper, therefore, it is omitted here.

Point 17: Section 5.2, the inertial parameters included in the dynamic model are missing and authors are invited to detail clearly the model.

Response 17: The authors sincerely thank the reviewer for the time, effort and suggestions spent in reviewing our manuscript. We have made correction according to the reviewer’s comments.

As shown in Figure 11, the dotted line is the joint rotation angle change curve, the variable is set to , the solid line is the joint torque change curve , and the variable is set to . From the torque formula , where is the distance from the centre of mass of the robot arm to the axis of the shoulder joint.

 is a transformation matrix between adjacent arm joints based on the D-H parameter model。

The maximum angle is  in the initial position, and returns to the original position

, and then gos to the maximum angle .

Point 18: Section 6, a dynamic torque sensor is used as mentioned on the block diagram. Authors are invited to detail the number of used torque sensors and way to use in control scheme.

Response 18: The authors sincerely thank the reviewer for the time, effort and suggestions spent in reviewing our manuscript. We have made correction according to the reviewer’s comments.

To realize the precise control of wearable upper limb rehabilitation robots, and reduce the motor torque error between the actual output torque and the theory, and let patients achieve the best effect of rehabilitation, the torque demand larger shoulder joint is designed in this paper. In addition, the drive motor of bending/stretching exercise and elbow flexion/stretch exercise drive motor installed on the dynamic torque sensor based on the basis of the fourth section results, that is to say, wearable upper limbs rehabilitation robot dynamics analysis results of output torque. The dynamic torque sensor transmits the information to the computer through the data acquisition card. The difference between the theoretical torque and the measured torque is input into the PID controller and then the torque of the motor is adjusted in real time to achieve the optimal control.

Point 19: Section 6, the cartesian trajectory of the end-effector is necessary in addition to a comparison between the test and simulation results.

Response 19: The authors sincerely thank the reviewer for the time, effort and suggestions spent in reviewing our manuscript. We have made correction according to the reviewer’s comments.

Since the end effector of the wearable upper limb rehabilitation robot uses a flexible parallel mechanism driven by a rope and supported by a compression spring, the cartesian trajectory of the end effector of the wearable upper limb rehabilitation robot is not studied in this paper. This part is the next research task in the future.

Finally, we would like to say thanks again sincerely to the editor and anonymous reviewers for their time and effort spent in handling our manuscript, as well as providing us many constructive comments for improving very much the presentation and quality of this manuscript. According to reviewers’ suggestions, we tried our best to improve the manuscript and made some changes in the manuscript. These changes will not influence the content and framework of the paper. We appreciate for Editors’ warm work earnestly and hope that the correction will meet with approval. Once again, thank you very much for your comments and suggestions.

Reviewer 3 Report

A major rewrite of the paper is necessary.

The designed structure is not wearable, it is fixed.
The text must be cleaned up, it has a lot of irrelevant phrases, for example 
"rehabilitation" occurs 184 times, "rehabilitation robot" 113 times. It really looks like hunting for high page count
Please improve on English language (for example it's Monte Carlo method not Monte carol)
Motor torques claims in text are not supported by fig. 13 (max. torque is <0.4 Nm).

Author Response

Dear Editors and Reviewers,

On behalf of all the contributing authors, I would like to express our sincere appreciation for your letter and reviewers’ constructive comments concerning our article entitled “Design and analysis of a wearable upper limb rehabilitation robot with characteristics of tension mechanism” (Manuscript No.: applsci-730801).

According to these comments, we have made extensive modifications to our manuscript and supplemented extra simulations to make our results convincing. Point-by-point responses to the comments of associate editor and reviewers are listed below with the corresponding revisions highlighted in blue colour in the revised manuscript.

Response to Reviewer 3 Comments

Point 1: The designed structure is not wearable, it is fixedï¼›The text must be cleaned up, it has a lot of irrelevant phrases, for example.

Response 1: The authors sincerely thank the reviewer for raising such a comment. We have made correction according to the reviewer’s comments.

The wearable upper limb rehabilitation robot designed in this article is wearable, which consists of a vertical wearable rehabilitation robot and a seat: the patient sits on the seat during the rehabilitation process; the patient's arm passes through the traction of the rehabilitation robot institutional contacts. The arm passively performs full-circle rotation by driving the curved rack and provides resistance during active movements of the shoulders. The adjustment mechanism of seat height can be adjusted according to the height of the human upper limb and the body shape, which adapt to different treatment environments and patient body differences, and ensure that the affected limb is trained on the sagittal plane during the rehabilitation process.

Figure 5. 3D model of wearable upper limb rehabilitation robot.

(a)                                             (b)

Figure 15. Test and control system diagram of rehabilitation robot. (a) Tester wear prototype test; (b) Physical map of control system.

To satisfy the rehabilitation needs of patients with limb disorders, a wearable upper limb rehabilitation robot is designed and developed in this article, which is mainly a device for mid-term semi-active rehabilitation training and post-active rehabilitation training for stroke patients. Owing to understanding the disadvantages of traditional rehabilitation training and the performances of rehabilitation robots, combined with the human upper limb muscle anatomy characteristic and relevant parameters, we determine the arm movement of each joint angle range from all the bones and joints of upper limb movement characteristics, this paper proposes a design scheme of tensegrity structure wearable upper limbs rehabilitation robot. The wearable upper limb rehabilitation robot is utilized to the exercise rehabilitation treatment of hemiplegic limb to maintain the range of motion of the limb, prevent the muscle atrophy of the limb, enhance the muscle strength of the limb, and promote the recovery of the limb function. Therefore, it can provide an effective rehabilitation equipment for patients with hemiplegia of upper limb caused by stroke.

Point 2: The text must be cleaned up, it has a lot of irrelevant phrases, for example

"rehabilitation" occurs 184 times, "rehabilitation robot" 113 times. It really looks like hunting for high page count.

Response 2: The authors sincerely thank the reviewer for such a helpful suggestion. We have made explanation according to the reviewer’s comments.

Point 3: Please improve on English language (for example it's Monte Carlo method not Monte carol).

Response 3: The authors sincerely thank the reviewer for the time, effort and suggestions spent in reviewing our manuscript. We have made correction according to the reviewer’s comments.

Point 4: Motor torques claims in text are not supported by fig. 13 (max. torque is <0.4 Nm).

Response 4: The authors sincerely thank the reviewer for such a helpful suggestion. We have made explanation according to the reviewer’s comments.

In this paper, the dynamic analysis uses the three-dimensional model established in Section 2 into Adams (Automatic dynamic analysis of mechanical systems) software, and then set the parameters of the imported model, and add driving to each joint of the wearable upper limb rehabilitation robot. Considering the differences in the joint bearing capacity of patients, the bearing value of most people is taken during initialization. The resistance of the affected limb to the rehabilitation robot is set to 100N in the horizontal movement direction and 200N in the vertical movement direction. Near the elbow joint, the torque of the shoulder joint is 4N m, and the force point is at the centre of mass of the forearm.

Finally, we would like to say thanks again sincerely to the editor and anonymous reviewers for their time and effort spent in handling our manuscript, as well as providing us many constructive comments for improving very much the presentation and quality of this manuscript. According to reviewers’ suggestions, we tried our best to improve the manuscript and made some changes in the manuscript. These changes will not influence the content and framework of the paper. We appreciate for Editors’ warm work earnestly and hope that the correction will meet with approval. Once again, thank you very much for your comments and suggestions.

Round 2

Reviewer 1 Report

The authors made a significant effort to improve the quality of the manuscript. I believe it can be published in the present form.